# ON GOOGLE'S LLM WATERMARKING SYSTEM: THEORETICAL ANALYSIS AND EMPIRICAL VALIDATION

## ABSTRACT

Google's SynthID-Text, the first ever production-ready generative watermark system for large language model, designs a novel Tournament-based method that achieves the state-of-the-art detectability for identifying AI-generated texts. The system's innovation lies in three key components: 1) a new Tournament sampling algorithm for watermarking embedding, 2) a detection strategy based on the introduced score function (e.g., Bayesian or mean score), and 3) a unified design that supports both distortionary and non-distortionary watermarking methods.

This paper presents the first theoretical analysis of SynthID-Text, with a focus on its detection performance and watermark robustness, complemented by empirical validation. For example, we prove that the mean score is inherently vulnerable to increased tournament layers, and design a *layer inflation attack* to break SynthID-Text. We also prove the Bayesian score offers improved watermark robustness w.r.t. layers and further establish that the optimal Bernoulli distribution for watermark detection is achieved when the parameter is set to 0.5. Together, these theoretical and empirical insights not only deepen our understanding of SynthID-Text, but also open new avenues for analyzing effective watermark removal strategies and designing robust watermarking techniques. Source code is available at `https://anonymous.4open.science/r/Break-Synth-ID-text-EE5D/`.

## 1 INTRODUCTION

As large language models (LLMs) (Zhang et al., 2022; Touvron et al., 2023; Bubeck et al., 2023) are increasingly integrated into real-world applications, the need for reliable mechanisms to identify AI-generated content has become more urgent. In the domain of text generation, LLMs have blurred the line between human- and machine-authored content (Köbis & Mossink, 2021; Clark et al., 2021; Jakesch et al., 2023). Given their widespread adoption in areas such as education, software development, and content creation, the ability to identify LLM-generated text is critical to ensuring safe and responsible use of this technology (Taori & Hashimoto, 2023; Shumailov et al., 2024).

Watermarking—a technique for embedding hidden and verifiable signals that can be detected later—is a promising solution for identifying AI-generated content. It can be applied at various stages of the text generation pipeline (Dathathri et al., 2024a; Wu et al., 2025): during the generation process itself (referred to as *generative watermarking*), by modifying already generated text, by altering the training data of the LLM, or via post-hoc detection. Among these approaches, generative watermarking (Aaronson, 2023; Kirchenbauer et al., 2023; Kuditipudi et al., 2024; Wouters, 2024; Dathathri et al., 2024a; Fu et al., 2024; Liu & Bu, 2024; Christ et al., 2024; Hu et al., 2024; Huo et al., 2024; Zhao et al., 2024; Fairoze et al., 2025; Liu et al., 2024; Hou et al., 2024; Zhang et al., 2024b)—which allows watermarks to be embedded during generation while carefully balancing text quality and computational efficiency—has emerged as the dominant focus in the field.

**SynthID-Text** (Dathathri et al., 2024a), recently developed by Google DeepMind, is the *first ever industrial-scale and production-ready* generative watermarking framework designed to be efficient, non-invasive, and detectable at scale. It represents a significant advancement in LLM watermarking—its deployment in Google's Gemini and Gemini Advanced (SynthID Team, 2024) demonstrates its viability in real-world systems. SynthID-Text builds upon prior generative watermarking approaches, but introduces a novel sampling algorithm called *Tournament Sampling* to generate the next token. At a high level, each layer of the tournament assigns a pseudo-random number (referred to as a

*g-value*) to every token in the vocabulary, reflecting the degree to which a token aligns with the watermarking signal at that layer. These *g*-values are layer-specific and are used to determine the winning token through a multi-round elimination strategy. The watermark is embedded by subtly biasing this tournament process in favor of tokens with stronger alignment signals. To enable watermark detection, SynthID-Text defines a *score function* that aggregates the *g*-values across all layers and tokens. If this score exceeds a (predefined or learnt) threshold, the corresponding text is classified as watermarked. *Detectability is measured by the true positive rate (TPR) at a small false positive rate (FPR)*, e.g., FPR=1%.

SynthID-Text has empirically shown to substantially outperform the best SOTA. For instance, it achieves a TPR=85% vs. SOTA 73% at FPR=1% (Figure 3(a) in Dathathri et al. (2024a)) when generating 1,500 watermarked and 10,000 unwatermarked texts with 400 token by the Gemma-7B LLM (Team et al., 2024) on the ELI5 dataset (Fan et al., 2019) under 30 tournament layers, a Bernoulli(0.5) *g*-value distribution and the Bayesian score function (see more details in Section 3); and has been successfully implemented and scaled to Google production systems.

*While SynthID-Text demonstrates strong empirical performance, its underlying detection mechanism and robustness have not yet been rigorously analyzed from a theoretical perspective*[1]. In this work, we present a formal analysis of SynthID-Text's detection performance (TPR@FPR) on the underlying *g*-value distribution (Bernoulli or Uniform) and score function (mean score or Bayesian score). Our theoretical analysis primarily utilizes the Central Limit Theorem, which allows us to derive closed-form expressions for the expected value and variance of the score function for the considered *g*-value distributions. These expressions enable the estimation of the expected TPR at a given FPR.

**Theoretical Findings:** Our theoretical findings are summarized below.

1. Under the mean score, the TPR at a fixed FPR is a *unimodal* function w.r.t. the number of tournament layers. That is, the TPR first increases and then decreases, regardless of the specific *g*-value distribution used. We also show the TPR will eventually be the FPR, as the layers grow.
2. Under the Bayesian score, the TPR (at a given FPR) is a *monotonically non-decreasing* function as the number of tournament layers increases, regardless of the used *g*-value distribution. We also theoretically show that the TPR will saturate beyond a layer number.
3. The optimal Bernoulli distribution to obtain the highest TPR at a fixed FPR is Bernoulli(0.5).

We also highlight that our theoretical findings 1 and 2 match SynthID-Text's empirical results as shown in Fig. C1 in the Supplement Material (Dathathri et al., 2024b).

**Implications:** Our theoretical findings also inspire certain implications.

1. **SynthID-Text with the mean score is vulnerable to watermark removal attacks.** We design a *black-box layer inflation attack* (see Figure 1 *Right*) by exploiting the unimodality property of the detection metric. Specifically, an attacker can simply concatenate the current SynthID-Text watermarked LLM with a copied instance, thereby artificially increasing the number of layers. Due to the unimodal behavior of the TPR under the mean score function, this increase in layers can eventually reduce the TPR, thus weakening the effectiveness of detection.
2. **SynthID-Text with the Bayesian score is more favorable, though more time-consuming and ultimately saturates.** Since the TPR under the Bayesian score is *non-decreasing*, it is generally more effective in practice with more layers. But we also emphasize that computing the Bayesian score incurs much higher computational cost compared to the mean score.
3. **Finding 3 suggests the default Bernoulli**(0.5) **used in SynthID-Text achieves optimality.**

## 2 RELATED WORK

Existing approaches for identifying AI-generated texts can be mainly summarized below (Dathathri et al., 2024a; Wu et al., 2025; Xuandong Zhao, 2025).

**Post-hoc detection-based approaches** (Mitchell et al., 2023; Verma et al., 2024; Hans et al., 2024; Krishna et al., 2023; Munyer & Zhong, 2023). These methods typically analyze statistical patterns—such as token frequencies, lexical entropy, or decoding structures—and then train a machine learning classifier to distinguish between human-written and AI-generated texts. Such techniques

---

[1]The focus on SynthID-Text is also consistent with prior work (Jovanović et al., 2024), which centers its robustness analysis on the KGW watermarking approach (Kirchenbauer et al., 2023).

offer broader detection capabilities without requiring intervention during the text generation process. Examples include DetectGPT (Mitchell et al., 2023), which leverages curvature in the log-likelihood space through global sampling, and paraphrase-invariant token statistics (Krishna et al., 2023). However, the practical effectiveness of these methods is limited by their inconsistent performance (Elkhatat et al., 2023)—they tend to underperform on out-of-domain data and exhibit elevated false positive rates for certain groups, such as non-native speakers (Liang et al., 2023). Moreover, these classifiers rely on detectable differences between human- and machine-generated text—differences that are likely to diminish as LLM capabilities continue to improve (Sadasivan et al., 2025; Zhang et al., 2024a). This trend necessitates ongoing classifier maintenance, including frequent retraining and recalibration, which can be computationally expensive and operationally burdensome.

**Watermarking-based approaches** embed hidden watermark signals into generated text that can be subsequently detected. These watermark signals can be introduced into the existing text (e.g., through rule-based transformations such as synonym substitution or the insertion of special Unicode characters (Kamaruddin et al., 2018)), into the training data (e.g., via specific trigger phrases), or during the generation process itself (commonly referred to as *generative watermarking*) (Dathathri et al., 2024a). The first two approaches often leave detectable artifacts in the text, reducing their stealthiness. As a result, generative watermarking has emerged as the dominant method. Generative watermarking (Aaronson, 2023; Kirchenbauer et al., 2023; Kuditipudi et al., 2024; Wouters, 2024; Dathathri et al., 2024a; Fu et al., 2024; Wang et al., 2024; Liu & Bu, 2024; Christ et al., 2024; Hu et al., 2024; Huo et al., 2024) generally falls into *non-distortionary* and *distortionary* ones.

*Non-distortionary approaches* (Aaronson, 2023; Fu et al., 2024; Kuditipudi et al., 2024; Christ et al., 2024; Hu et al., 2024) preserve text quality by embedding watermark signals without modifying the output token distribution–the distribution of the token outputted by watermarked LLMs equals to the original LLM distribution. These techniques base on, e.g., Gumbel sampling (Aaronson, 2023; Fu et al., 2024), and the embedded watermark is stealthy. *Distortionary approaches* (Kirchenbauer et al., 2023; Wouters, 2024; Liu & Bu, 2024; Huo et al., 2024), in contrast, deliberately bias token distribution to improve watermark detectability, but at the cost of text quality loss. The pioneer method (Kirchenbauer et al., 2023) carefully partitions the vocabulary into a green list and a red list based on a secret key, so as to increase the sampling likelihood of green tokens during decoding. Follow-up works mainly enhance text quality via gradient-based token reweighting (Huo et al., 2024), minimizing perplexity of generated texts (Wouters, 2024), and injecting watermark (Liu & Bu, 2024) to token distributions with high entropy while keeping low-entropy token distributions untouched.

**SynthID-Text** (Dathathri et al., 2024a) unifies both non-distortionary and distortionary watermarking approaches through a novel mechanism called *Tournament Sampling*. Rather than selecting the next token solely based on top-ranked probabilities, SynthID-Text conducts a multi-layer tournament consisting of pairwise token comparisons. This approach enables flexible watermark embedding while preserving control over generation quality. Notably, SynthID-Text also outperforms existing watermarking methods across a range of evaluation benchmarks.

*In this paper, we focus mainly on the non-distortionary version of SynthID-Text because it reflects the most practical setting, where watermarking must preserve output quality and semantic fidelity. Due to the noticeable quality degradation introduced by distortionary watermarking, such methods may not be practical for real-world deployment where maintaining high text quality is essential. Moreover, the current version of Google's SynthID-Text operates in the non-distortionary setting, and all official results are based on this configuration.*

## 3 BACKGROUND

### 3.1 SYNTHID-TEXT FOR GENERATIVE LLM WATERMARKING

An LLM is a probabilistic model trained to estimate the likelihood of a sequence of tokens. Given a vocabulary $\mathcal{V}$ and a context sequence $x_{<t} = (x_1, x_2, \ldots, x_{t-1})$ consisting of $t-1$ tokens from $\mathcal{V}$, an LLM $p_{LM}(\cdot \mid x_{<t})$ defines a probability distribution over the next token $x_t \in \mathcal{V}$ given the preceding text $x_{<t}$. An LLM typically uses a sampling algorithm (e.g., greedy, top-$k$, top-$p$, or temperature sampling) to draw the next token $x_t$ from $p_{LM}(x_t \mid x_{<t})$. This process is repeated iteratively to generate an output sequence $x = (x_1, x_2, \ldots, x_T)$ of $T$ tokens.

**SynthID-Text** embeds watermarks during the token generation phase of LLMs without altering the model architecture. The core idea is to *bias* the sampling distribution in a subtle way using *Tournament*

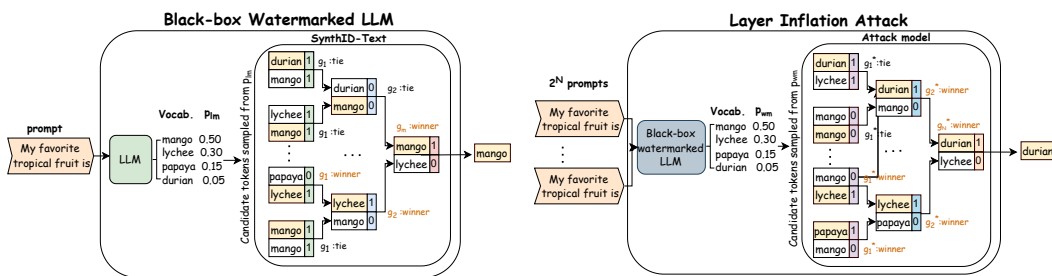

Figure 1: *Left*: Overview of non-distortionary SynthID-Text's Tournament-based watermarking with $m$ layers; *Right*: Layer inflation attack by appending the SynthID-Text's Tournament sampling with $N$ layers to remove the watermark.

*Sampling*, introducing statistical correlations that can later be detected. The watermarking process of non-distortionary SynthID-Text involves the below primary components:

1. **Random Seed Generator:** At each token generation step $t$, produce a seed $r_t$ that depends on the recent $H$ tokens, a hash function $h$, and a secret key $k$, i.e., $r_t = h(x_{t-H}, \ldots, x_{t-1}, k)$, where $h$ is deterministic given $x_{<t}$ and $k$, but unpredictable without $k$.
2. **Tournament Sampling**: A multi-layer (say, $m$-layer) tournament is used to sample the next token $x_t$. With a random seed $r_t$, pseudorandom $g$-*value watermarking functions* $\{g_\ell(x_t, r_t)\}$ of all $m$ layers for any candidate token $x_t \in \mathcal{V}$ are computed. Note that the hash function generates outputs that are statistically indistinguishable from IID Bernoulli samples, and hence the resulting $g$-values are independent. *For conciseness, we will write $g_{t,\ell} = g_\ell(x_t, r_t)$ to refer to $g$-values.* Tournament sampling is constructed to favor tokens from the LLM distribution that are expected to attain higher values under the $g$-value functions.
   - Sample $m' = 2^m$ candidate tokens from $p_{LM}(\cdot \mid x_{<t})$, possibly with repeats.
   - For each layer $\ell = 1, \ldots, m$, $g_\ell(\bar{x}, r_t)$ assigns a score to each candidate token $\bar{x}$ given seed $r_t$.
   - Run a knockout-style tournament among these $m'$ candidates: (a) In layer 1: group into $m'/2$ pairs, and in each pair pick the token with higher $g_1(\cdot, r_t)^2$; (b) In layer 2: regroup winners into pairs, use $g_2$; and (c) Continue until a single token remains after layer $m$; that becomes $x_t$.

   *Figure 1 Left briefly shows how a tournament process with $m$ layers chooses the next token, where tokens are split into pairs of competing tokens in the non-distortionary setting.*
3. **Watermark Detection (via a Score Function):** To detect whether a outputted text $x = (x_1, x_2, \ldots, x_T)$ is watermarked, a *score function* $\text{Score}(x)$ measuring how highly $x$ scores with respect to $\{g_\ell\}$ is utilized. *Under watermarking, this score is higher than in unwatermarked or random text, since sampling favored tokens with higher $g$ values.* If $\text{Score}(x)$ exceeds a threshold (e.g., $\tau$), $x$ is identified as watermarked ($w$); otherwise, it is considered unwatermarked ($\neg w$).

### 3.2 $g$-VALUE FUNCTION/DISTRIBUTION, SCORE FUNCTION, AND DETECTION METRIC

**$g$-Value Function/Distribution:** Tournament sampling requires $g$-values to decide which tokens win each match in the tournament. SynthID-Text uses the below two $g$-value distributions:

- **Bernoulli(0.5)**: a Bernoulli distribution with parameter $0.5$. It means the $g$-value random function $g_{t,\ell} \sim \text{Bernoulli}(0.5)$ takes the value 0 or 1 with probability $0.5$:
- **Uniform(0, 1)**: a uniform distribution over the interval $[0, 1]$. It means $g$-value random function $g_{t,\ell} \sim \text{Uniform}(0, 1)$ takes values between 0 and 1 with equal probability density.

**Score Function** $\text{Score}(x)$: It is used for distinguishing between watermarked and non-watermarked texts and plays a central role in SynthID-text's detectability. SynthID-Text uses below score functions:
- **Mean Score (MS):** the mean score function across all tokens and layers is defined as

$$\text{MS}(x) = \frac{1}{Tm} \sum_{t=1}^{T} \sum_{\ell=1}^{m} g_\ell(x_t, r_t) \tag{1}$$

When $g$-values are sampled from Bernoulli(0.5) or a Uniform(0,1), the MS of a text falls within [0,1]. For unwatermarked text, the expected MS is 0.5, while watermarked text tends to have a higher MS.

---

[2]Ties are resolved by randomly selecting one of the tied tokens with equal probability 0.5.

- **Bayesian Score (BS):** the Bayesian approach in SynthID-Text formulates watermark detection as a binary hypothesis test: watermarked ($w$) or not ($\neg w$). Given the observed $g$-values $\{g_{t,\ell}\}_{1 \leq t \leq T, 1 \leq l \leq m}$ as observed evidence. The goal is to estimate the posterior probability that the text is watermarked ($w$), denoted by $P(w \mid g)$. This estimation is based on the prior probability $P(w)$ of a text $x$ being watermarked, along with the likelihoods $P(g \mid w)$ and $P(g \mid \neg w)$, which represent the distributions of $g$-values under the watermarked and unwatermarked hypotheses, respectively.

To obtain a score between 0 and 1, the sigmoid function $\sigma(\cdot)$ is applied to the log posterior odds:

$$\log \frac{P(w \mid g)}{P(\neg w \mid g)} = \log \frac{P(g \mid w)}{P(g \mid \neg w)} + \log \frac{P(w)}{1 - P(w)} \tag{2}$$

This value, referred as the **Bayesian score**, quantifies the probability that a given text is watermarked:

$$\text{BS}(x) = \sigma\left(\log P(g \mid w) - \log P(g \mid \neg w) + \log P(w) - \log(1 - P(w))\right), \tag{3}$$

where the prior $P(w)$ is estimated from labeled data and $P(g \mid w)$ and $P(g \mid \neg w)$ can be calculated from the (unwatermarked and watermarked) $g$-value distribution (see below Theorem 2).

**Detection Metric ($TPR@FPR = \epsilon$):** Watermarking methods need to define a detection metric to distinguish between watermarked and unwatermarked texts. The detection metric used in SynthID-Text is the *True Positive Rate (TPR) at a small False Positive Rate (FPR)*, e.g., computing the TPR with FPR=1% is used to measure the watermark detectability of SynthID-Text.

## 3.3 PRELIMINARIES

In this section, we present key definitions and theorems that underpin our theoretical results. Without loss of generality, we assume that SynthID-Text has $m$ tournament layers, and the LLM generates $T$ tokens in total. A statistical framework (Li et al., 2025) for formulating watermarks is in Appendix A.

**Definition 1** (True Positive Rate (TPR) (Van Trees, 2004)). *Given the probability distribution of the watermarked distribution $p_w(x)$, the TPR no smaller than $\tau$ is defined as:*

$$\mathbb{E}[TPR(\tau)] = p_w(x \geq \tau) = 1 - CDF_w(\tau), \tag{4}$$

*where $CDF_w(\tau)$ is the CDF (cumulative distribution function) of the watermarked text. Here, TPR is a random variable because it depends on a distribution over inputs.*

**Definition 2** (Collision probabilities Dathathri et al. (2024a)). *Given a probability distribution $p$, the collision probability $C_p$ of $p$ is the probability that two samples drawn independent and identically distributed (i.i.d.) from $p$ are the same. If $p = (p_1, p_2, \cdots, p_N)$ is discrete, $C_p = \sum_{i=1}^{N} p_i^2$.*

The below theorem describes the watermarked $g$-value distribution, denoted as $f_{gw}$, in terms of the unwatermarked $g$-value distribution $f_g$ and the collision probabilities $\{C_{\ell,t}\}$.

**Theorem 1** (Watermarked $g$-value distribution for single-layer tournament (Dathathri et al. (2024a)) ). *Let $C_{l,t}$ be the collision probability w.r.t layer $l$ at $t$-th token and $F_g$ the CDF of the unwatermarked $g$-value distribution $f_g$. The CDF $F_{gw}$ of the watermarked g-value distribution $f_{gw}$ is given by:*

$$F_{gw}(g_{t,\ell}) = C_{t,\ell} F_g(g_{t,\ell}) + (1 - C_{t,\ell}) F_g(g_{t,\ell})^2, \tag{5}$$

*where if $g$ is continuous, the PDF $f_{gw}$ is given by:*

$$f_{gw}(g_{t,\ell}) = f_g(g_{t,\ell}) \left[C_{t,\ell} + 2(1 - C_{t,\ell}) F_g(g_{t,\ell})\right], \tag{6}$$

*and if $g$ is discrete, the PMF $f_{gw}$ is given by:*

$$f_{gw}(g_{t,\ell}) = f_g(g_{t,\ell}) \left[C_{t,\ell} + (1 - C_{t,\ell})(2F_g(g_{t,\ell}) - f_g(g_{t,\ell})\right] \tag{7}$$

**Theorem 2** (Bayesian likelihoods for $m$-layer Tournament sampling (Dathathri et al., 2024a)). *For $m$-layer Tournament sampling, the likelihoods $P(g|w)$ and $P(g|\neg w)$ can be factorized as:*

$$P(g|\neg w) = \prod_{t=1}^{T} \prod_{\ell=1}^{m} f_g(g_{t,\ell}), \ \ P(g|w) = \prod_{t=1}^{T} \prod_{\ell=1}^{m} \sum_{c=1}^{2} P(g_{t,\ell}|\psi_{t,\ell} = c) P(\psi_{t,\ell} = c|g_{t,<\ell}) \tag{8}$$

*where $\psi_{t,\ell}$ is a random variable representing the number of unique tokens on layer $\ell$, at timestep $t$. Furthermore, $P(g_{t,\ell}|\psi_{t,\ell} = c)$ can be written in terms of the $g$-value distribution $f_g$ and $F_g$ as:*

$$P(g_{t,\ell}|\psi_{t,\ell} = c) = \begin{cases} c F_g(g_{t,\ell})^{c-1} f_g(g_{t,\ell}) & \text{if } f_g \text{ is continuous} \\ F_g(g_{t,\ell})^c - [F_g(g_{t,\ell}) - f_g(g_{t,\ell})]^c & \text{if } f_g \text{ is discrete} \end{cases} \tag{9}$$

Plugging Equations 8 and 9 into Equation 3, one can rewrite Bayesian score as the function of the sum of $g$-values of all tokens and layers.

# 4 THEORETICAL ANALYSIS

In this section, we present the theoretical analysis of SynthID-Text's detection performance (TPR@FPR) with respect to the $g$-value function and score function. The detailed procedure is as follow: 1) Analyze the behavior of score function; 2) State a general form of TPR at a FPR; and 3) Show the trend of TPR at a FPR w.r.t. the tournament layers. We respectively show the theoretical analysis on the Mean Score and Bayesian Score. **All proofs are deferred to Appendix.**

## 4.1 MEAN SCORE

**Score Function Analysis.** In the MS function, a large number of $mT$ random variables are summed—specifically, the random $g$-value of $m$ layers and $T$ tokens—and the distribution of their sum can be approximated by a normal distribution according to the Central Limit Theorem (CLT, with proof in Appendix C). With it, the probability distribution of the watermarked MS function can be defined as:

$$\text{MS}(x) \sim \text{Normal}\left(\mathbb{E}[MS(x)], \text{Var}[MS(x)]\right) \tag{10}$$

**Derive TPR at FPR.** Next, we state a general form of TPR@FPR under normally distributed MS.

**Proposition 1** (TPR@FPR $= \epsilon$ for normally distributed MS)**.** *Given a FPR $= \epsilon$, the expected TPR@FPR$= \epsilon$ can be defined as:*

$$\mathbb{E}[TPR(\tau(\epsilon))|FPR = \epsilon] = 1 - \Phi\left(\frac{\tau(\epsilon) - \mathbb{E}[MS(x)|w]}{\sqrt{Var[MS(x)|w]}}\right), \tag{11}$$

where $\Phi$ is the CDF of the Normal distribution, $\tau(\epsilon)$ is detection threshold, and $\mathbb{E}[MS(x)|w]$ and $\text{Var}[MS(x)|w]$ are the expected value and variance of the watermarked MS, respectively.

The results of calculating $\tau(\epsilon)$, $\mathbb{E}[MS(x)|w]$, and $\text{Var}[MS(x)|w]$ with different $g$-value distributions are stated in below theorems.

**Theorem 3.** *For the Bernoulli(0.5) $g$-value distribution, the expected value and variance of MS conditioned on output $x$ being watermarked are given by:*

$$\mathbb{E}[MS(x)|w] = \frac{1}{mT}\sum_{t,\ell} p_{t,\ell} = \frac{1}{mT}\sum_{t,\ell}\frac{3 - C_{t,\ell}}{4}, \quad Var[MS(x)|w] = \left(\frac{1}{mT}\right)^2 \sum_{t,\ell} p_{t,\ell}(1 - p_{t,\ell})$$

**Theorem 4.** *For the Uniform(0,1) $g$-value distribution, the expected value and variance of MS conditioned on output $x$ being watermarked are given by:*

$$\mathbb{E}[MS(x)|w] = \frac{1}{mT}\sum_{t,\ell} p_{t,\ell} = \frac{1}{mT}\sum_{t,\ell}\frac{4 - C_{t,\ell}}{6}, \quad Var[MS(x)|w] = \left(\frac{1}{mT}\right)^2 \sum_{t,\ell} \left[p_{t,\ell}(1 - p_{t,\ell}) - \frac{1}{6}\right]$$

*Note that, in the above two theorems, the expected MS of watermarked text is larger than that of unwatermarked text, which is 0.5.*

**Theorem 5.** *For the Bernoulli(0.5) $g$-value distribution and FPR$=\epsilon$, $\tau(\epsilon) = \frac{1}{2} + \frac{\Phi^{-1}(1-\epsilon)}{2\sqrt{mT}}$.*

**Theorem 6.** *For the Uniform(0,1) $g$-value distribution and FPR$=\epsilon$, $\tau(\epsilon) = \frac{1}{2} + \frac{\Phi^{-1}(1-\epsilon)}{\sqrt{12mT}}$.*

**TPR Trend Analysis.** We finally analyze the trend of detection metric (TPR@FPR) w.r.t. w.r.t. the tournament layers $m$, with the theorem stated below:

**Theorem 7.** *With the $g$-value distribution being Bernoulli(0.5) and Uniform(0,1), we have*

$$\mathbb{E}[TPR(\tau(\epsilon))|FPR = \epsilon] = \begin{cases} 1 - \Phi\left(\dfrac{\hat{A}\sqrt{m} - Am}{B\sqrt{m}}\right) & \text{if } m < M \\ 1 - \Phi\left(\dfrac{\hat{A}\sqrt{m} - \hat{B}}{\sqrt{\hat{C}m - \hat{D}}}\right) & \text{if } m \geq M \end{cases} \tag{12}$$

*Where $\hat{A}, A, \hat{B}, B, \hat{C}, \hat{D} > 0$ are different in Bernoulli(0.5) and Uniform(0,1) and **their detailed forms are deferred to Appendix C.7**. Additionally, M is, for the first time, the layer number after which $C_{M,t} = 1$ (i.e., when the collision probability becomes 1) for all tokens $t$.*

**Corollary 1.** *The expected TPR in Equation 12 is a unimodal function—it first increases and then decreases as $m$ grows.*

**Corollary 2.** *The peak TPR is at the $M$-th layer. The TPR will decrease to be a saturate value $\epsilon$.*

**Intuitive Explanation:** After a sufficient number of tournament layers, the expected value of the detection statistic converges to a stable value. However, the variance of the g-values continues to increase, since each additional layer effectively introduces new random variables, and the cumulative variance grows with every addition. As a result, the watermarked and unwatermarked distributions gradually overlap, reducing their separability and thereby decreasing detectability.

### 4.2 BAYESIAN SCORE

In this section, the detectability under the Bayesian Score is analyzed.

**Score Function Analysis.** Let $X = \frac{P(g|w)}{p(g|\neg w)}$ and $\alpha = \frac{P(w)}{P(\neg w)}$, we can simplify BS (Equation 3) as:

$$\text{BS}(x) = \sigma[\log(\alpha X)] = \frac{1}{1 + e^{-\log(\alpha X)}} = \frac{\alpha X}{\alpha X + 1} \tag{13}$$

Further, if we define the random variable $X_{t,\ell} = \log\left(\frac{f_{gw}(g_{t,\ell})}{f_g(g_{t,\ell})}\right)$, then $X$ is a log-normal distribution based on the CLT (proof in Appendix D.1). That is,

$$X \sim \text{LogNormal}\left(\mathbb{E}[\sum_{t,\ell} \log(X_{t,\ell})], \text{Var}(\sum_{t,\ell} \log(X_{t,\ell}))\right). \tag{14}$$

**Derive TPR at FPR.** We state a general form of TPR@FPR for the BS below based on CLT:

**Proposition 2** (TPR@FPR = $\epsilon$ for the BS). *Given a FPR, the expected TPR@FPR= $\epsilon$ is defined as:*

$$\mathbb{E}[TPR(\tau(\epsilon))|FPR = \epsilon] = 1 - CDF_{BS|w}(\tau(\epsilon)) \tag{15}$$

*where the CDF of Bayesian Score based on the Central Limit Theorem is given by:*

$$CDF_{BS|w}(x) = \Phi\left(\frac{\ln\left(\frac{x}{\alpha(1-x)}\right) - \mathbb{E}[BS(x)|w]}{\sqrt{Var[BS(x)|w]}}\right), \tag{16}$$

*where $\alpha = \frac{P(\omega)}{P(\neg\omega)}$, $P(\omega)$ and $P(\neg\omega)$ are the ratio of watermarked samples and non-watermarked samples in training data; $\mathbb{E}[BS(x)|w]$ and $Var[BS(x)|w]$ are defined below.*

**Theorem 8** (Detailed form and proof in Appendix D.3). *With a Bernoulli(0.5) g-value distribution, the expected value and variance of the Bayesian score conditioned on $x$ are given by:*

$$\mathbb{E}[BS(x)] = \sum_{\ell,t} \mathbb{E}\left[\log\left(\hat{C}_{t,\ell} + (0.5 + g_{t,\ell})(1 - \hat{C}_{t,\ell})\right)\right] \tag{17}$$

$$Var[BS(x)] = \sum_{\ell,t} Var\left[\log\left(\hat{C}_{t,\ell} + (0.5 + g_{t,\ell})(1 - \hat{C}_{t,\ell})\right)\right] \tag{18}$$

*where $g_{t,\ell} \sim f_{gw}$ defined in Theorem 1 is for watermarked $x$, $BS(x)|w$; and $g_{t,\ell} \sim f_g$ for unwatermarked $x$, $BS(x)|\neg w$. $\hat{C}_{t,\ell}$ is the collision probability of train data where the Bayesian score was trained. The details of calculating $\hat{C}_{t,\ell}$ can be seen in Dathathri et al. (2024a).*

**Theorem 9** (Detailed form and proof in Appendix D.4). *With a g-value distribution Uniform(0,1), the expected value and variance of the Bayesian score conditioned on $x$ are given by:*

$$\mathbb{E}[BS(x)] = \sum_{\ell,t} \mathbb{E}\left[\log\left(\hat{C}_{t,\ell} + 2(1 - \hat{C}_{t,\ell})g_{t,\ell}\right)\right], \quad Var[BS(x)] = \sum_{\ell,t} Var\left[\log\left(\hat{C}_{t,\ell} + 2(1 - \hat{C}_{t,\ell})g_{t,\ell}\right)\right] \tag{19}$$

*Similarly, $g_{t,\ell} \sim f_{gw}$ is for watermarked $x$; and $g_{t,\ell} \sim f_g$ for unwatermarked $x$.*

**Theorem 10.** *Given a FPR=$\epsilon$, the detection threshold for the Bayesian score is given by:*

$$\tau(\epsilon) = 1 - \frac{1}{1 + \alpha \exp\left(\mathbb{E}[BS(x)|\neg w] + \Phi^{-1}(1 - \epsilon)\sqrt{\text{Var}(\mathbb{E}[BS(x)|\neg w])}\right)} \tag{20}$$

*where $\Phi^{-1}$ is the inverse of the Normal CDF. $\mathbb{E}[BS(x)|\neg w]$ and $Var[BS(x)|\neg w]$ are defined for unwatermarked $x$ in Theorems 8 and 9 for the Bernoulli(0.5) and Uniform(0,1) distribution, respectively.*

**TPR Trend Analysis.** The below corollaries show the trend of TPR w.r.t. the tournament layers $m$.

**Theorem 11.** *With the g-value distribution being Bernoulli(0.5) and Uniform(0,1), we have*

$$\mathbb{E}\left[TPR(\tau(\epsilon))|FPR = \epsilon\right] = 1 - \Phi\left(\frac{\mathbb{E}[BS(x)|\neg w] + \Phi^{-1}(1-\epsilon)\sqrt{\text{Var}[BS(x)|\neg w]} - \mathbb{E}[BS(x)|w]}{\sqrt{\text{Var}[BS(x)|w]}}\right) \quad (21)$$

*where the expectation and variance of BS for watermarked and unwatermarked texts are defined in the respective g-value distribution in Theorem 8 and Theorem 9.*

**Corollary 3.** *The expected TPR is a monotonically non-decreasing function, i.e., the TPR is non-decreasing as $m$ grows for both the Bernoulli(0.5) and Uniform(0,1) g-value distributions.*

**Corollary 4.** *TPR will saturate at the layer number $m$ when $\hat{C}_{m,t} = 1$ for the first time.*

**Intuitive Explanation:** Bayesian score leverages the exact distribution of g-values at each layer rather than their aggregated variance. This allows it to incorporate layer-wise distributional evidence when evaluating the hypothesis test, thereby improving its ability to reject the null hypothesis and maintaining higher detectability even as the number of layers increases.

## 5 EMPIRICAL EVALUATIONS

Following SynthID-Text, we conduct experiments on the ELI5 dataset using 1,000 texts, each with 100 tokens and setting FPR=1%. We use SynthID-Text's public implementation: LLM is Gemma-7B, $m$=30 by default, Temperature = 1.0, as well as two additional models GPT-2B and Mistral-7B.

### 5.1 EMPIRICAL VALIDATION FOR OUR TPR TREND

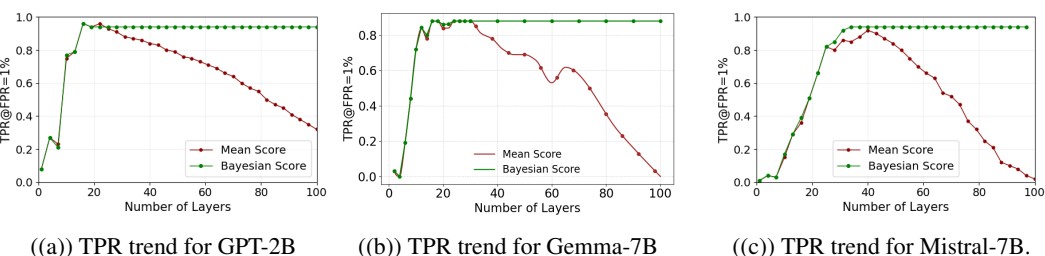

((a)) TPR trend for GPT-2B    ((b)) TPR trend for Gemma-7B    ((c)) TPR trend for Mistral-7B.

Figure 2: (a)-(c) show the TPR Trend on GPT-2B, Gemma-7B, and Mistral-7B, respectively.

In this experiment, we empirically verify the trend by our derived theoretical TPR at FPR = 1%. Our results for MeanScore (MS) and BayesianScore (BS) on the three models are shown in Figure 2. The results demonstrate the below observations:

- With MS, **TPR initially increases** (e.g., from 0.04 to 0.88 in Gemma-7B), as $m$ increases (e.g., from 1 to 28 on Gemma-7B). **TPR then decreases steadily**, eventually reaching a relatively low value, e.g., TPR=1% at l00 layers on Gemma-7B.

- With BS, **TPR increases and finally saturates.** Here, the Bayesian detector trains on $g$-values from watermarked and unwatermarked outputs, computing $P(w \mid g)$ by summing log-likelihood ratios across tokens/layers. Optimization minimizes cross-entropy loss between posterior predictions and true labels, with L2 regularization on likelihood parameters.

These empirical TPR trends well align with our theoretical TPR analysis in Corollaries 1- 4.

### 5.2 EMPIRICAL VALIDITY OF THE CLT ASSUMPTION

The correctness of our theoretical analysis relies on the CLT being applicable, which typically requires texts of moderate length. For short texts, the CLT assumption may not hold, and theoretical TPR may deviate from observed TPR results. We emphasize that, all existing LLM watermarking methods exhibit poor watermark detection performance on short texts. For example, the SOTA SynthID-Text

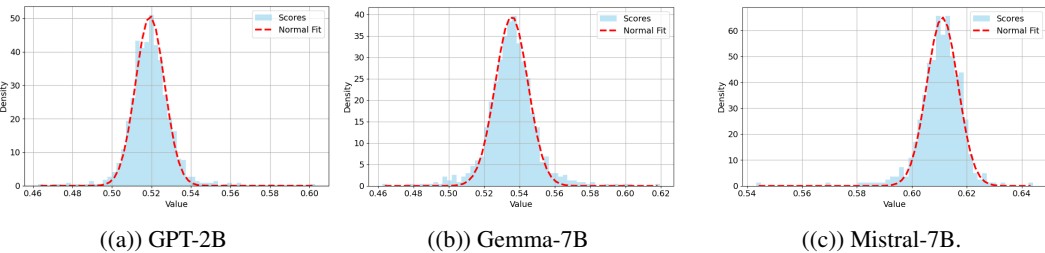

((a)) GPT-2B          ((b)) Gemma-7B          ((c)) Mistral-7B.

Figure 3: Gaussian distribution fitting of mean scores on the three models.

achieves a maximum TPR around 0.3 at FPR = 1%, when the text length is only 50 tokens, as shown in Fig. 1 and Extended Data in Dathathri et al. (2024a).

To further validate the CLT assumption, we apply the Anderson–Darling test to the distribution of mean scores across the 1,000 test samples and 30 layers, and Figure 3 shows the visualization of the distribution on the three models. We observe that the data passes the normality test, supporting the validity of the Gaussian assumption.

## 5.3 LAYER INFLATION ATTACK

Corollary 1 states the TPR with the mean score is a unimodal function and decreases when the layer number reaches a certain value, also validated in Figure 2. An attack can exploit such unimodality property to break SynthID-Text. Specifically, the attacker can simply append an extra (copied) SynthID-Text watermarked LLM to the current one (with black-box access) by artificially inflating the layer number. This will eventually reduce the TPR, thus weakening the watermark detection performance. Our *layer inflation attack* is detailed as follow:

Given an input prompt (e.g., *"My favorite tropical fruit is"*) and an LLM, SynthID-Text applies an $m$-layer tournament sampling over the token probabilities generated by the LLM to produce a winner token (e.g., *"mango"*). Our attack appends an additional $N$-layer tournament on top of this process. The steps are as follows (also see Figure 1 *Right*):

1. Feed the same prompt $2^N$ times to the LLM + SynthID-Text, yielding $2^N$ winner tokens.
2. Apply an additional $N$-layer tournament to these tokens to select a final winner (e.g., *"durian"*). This winner token is treated as the attack output.
3. Compute the mean score of the final token via Equation (1).

**Empirical detectability results:** We evaluate our attack using 1,000 known watermarked prompts sampled from ELI5. These prompts are all *correctly identified as watermarked* by SynthID-Text.

Table 1: Results of layer inflation attack

|      | GPT-2B | Gemma-7B | Mistral-7B |
|------|--------|----------|------------|
| TPR  | 0.05   | 0.00     | 0.01       |

Table 1 shows the detectability result after applying our attack with 15 additional layers. We can see that the TPR is very low. For instance, TPR=0 on Gemma-7B means that **all** watermarked prompts were *misclassified as unwatermarked*. Furthermore, we calculate that, before attack, the mean scores (their average is 0.548) of all test prompts are *above* the detection threshold of 0.515, calibrated for FPR = 1%. However, the mean scores (their average is 0.486) of all test prompts were *below* the detection threshold of 0.515 after our attack.

## 6 DISCUSSIONS, LIMITATIONS AND FUTURE WORK

**Mean Score vs. Bayesian Score in SynthID-Text.** The theoretical results presented in this work highlight critical differences between MS and BS used for watermark detection. Specifically, *MS is inherently vulnerable to watermark removal attacks*, which has been validated by our proposed layer inflation attack. Hence, while computationally simple and intuitive, MS lacks robustness in practical scenarios. In contrast, *BS could be favorable.* Though BS is computationally more expensive, it could be more effective and robust due to several reasons. First, Corollaries 3 and 4 show that the TPR is monotonically increasing with respect to the tournament layers, eventually saturating at a theoretical maximum as the watermark signal strengthens. This implies that SynthID-Text can use more layers in practice for better detection performance. Second, BS leverages prior knowledge

of both watermarked and unwatermarked training texts. Such knowledge is useful for identifying watermarked test texts in a more accurate and resilient fashion.

**Optimal $g$-value distribution in SynthID-Text.** Based on our theoretical analysis, we find that the optimal distribution for discrete $g$-values is the Bernoulli(0.5) distribution, stated below.

**Theorem 12** (Optimal Bernoulli Distribution for $g$-values). *Bernoulli$(0.5)$ achieves the highest TPR at a given FPR among all Bernoulli $g$-value distributions for Mean Score.*

This is because Bernoulli(0.5) maximizes the difference between expected value of the watermark and unwatermark signal, resulting in the largest separation between watermarked and unwatermarked distributions. This separation leads to the most confident watermark detection—maximally reducing the FPR while maintaining the highest TPR.

**Difference with Fernandez et al. (2023) on the CLT Assumption.** There is a key distinction between our work and Fernandez et al. (2023) on the CLT assumption: Fernandez et al. (2023) critiques existing LLM watermarking methods such as Aaronson (2023); Kirchenbauer et al. (2023), that rely on a *Z-test-based empirical FPR estimation*, assuming a CLT-based distribution of the test statistic. They show that this assumption can result in substantial deviation from the true *theoretical* FPR, and propose novel *non-asymptotic* statistical tests to more tightly control the empirical FPR. Our work, by contrast, aims to *theoretically analyze the expected TPR* at a *given theoretical FPR*, under the assumption that the score function–the sum over tokens and layers—is approximately Normal via CLT. Importantly, our analysis is independent of how the FPR is computed in practice. In other words, to empirically validate our expected TPR, one may use FPR computed via existing methods or more refined techniques proposed in Fernandez et al. (2023). Our empirical result in Section 5.2 also validates that the CLT assumption in this paper is reasonable with moderate text length.

**How Do the Findings Generalize to Other Watermarking Schemes?** Our theoretical findings extend to a broader class of watermarking schemes through a key property we refer to as *self-robustness*. A watermarking method is said to be self-robust if repeatedly applying its own watermarking procedure, i.e., stacking multiple watermark layers enhances detectability rather than diminishing it.

Our analysis shows that SynthID-Text under the MeanScore violates this property: as additional tournament layers are introduced, the statistical separation between watermarked and unwatermarked text progressively decreases, leading to weakened detectability. This reveals a broader vulnerability: any watermarking scheme whose detection relies on aggregated mean statistics is potentially susceptible to the same failure mode. We therefore argue that self-robustness should be considered a necessary design principle for future LLM watermarking systems.

**Limitations and Future Work.** This paper primarily focuses on the theoretical analysis of the non-distortionary setting of SynthID-Text, based on the fact that it preserves text quality. We outline several directions for future work:

1. *Extension to distortionary settings:* We plan to generalize our theoretical framework to analyze SynthID-Text under the distortionary watermarking setting.

2. *Theoretical comparison with prior work:* While SynthID-Text has empirically outperformed previous watermarking methods in detection performance under TPR@FPR, we aim to establish a comprehensive theoretical comparison with existing approaches.

3. *Robustness:* Like many existing LLM watermarking methods, SynthID-Text exhibits moderate performance degradation under adversarial scenarios such as paraphrasing attacks (Krishna et al., 2023; Jakesch et al., 2023). We plan to strengthen SynthID-Text with provable robustness guarantees against such attacks.

## 7 CONCLUSION

We present the first in-depth theoretical analysis of SynthID-Text, the first-ever effective, scalable, and production-ready LLM watermarking system developed and deployed by Google. Our analysis reveals that SynthID-Text with the mean score function, is fundamentally vulnerable to watermark removal attacks. In contrast, the Bayesian score offers improved robustness and detection effectiveness. Further, we prove that the use of a Bernoulli$(0.5)$ distribution for generating $g$-values is theoretically optimal for watermark detection. These findings lay the groundwork for future research on the theoretical foundations of LLM watermarking methods, including detection performance comparisons and provable robustness guarantees against watermark removal attacks.

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

# Appendix

## A   WORKING HYPOTHESIS

The question of whether a watermark is embedded in a given text sequence can be formulated as a hypothesis testing problem:

$H_0 : x_{1:n}$ is generated by an unwatermarked LLM, $H_1 : x_{1:n}$ is generated by a watermarked LLM.

In the SynthID-Text framework, this hypothesis testing problem involves the use of pseudorandom variables derived from internal model components. A general way to define the watermark feature $g_{t,\ell}$, consistent with existing constructions, is:

$$g_{t,\ell} = \mathcal{G}(x, r, \ell) = F_g^{-1}\Big(\frac{h(x, r, \ell)}{2^{n_{sec}}}\Big)$$

where $r$ is a secret key provided to the verifier, $F_g^{-1}$ is the generalized inverse CDF associated with $F_g$, and $h$ is a cryptographic hash function that takes the input text $x$, seed $r$, and layer index $\ell$ as input and returns a uniformly distributed $n$-bit integer. Dividing by $2^{n_{\text{sec}}}$ yields a value in $[0, 1]$, which converges to a uniform random variable for large $n$. Then inverse transform sampling is performed to turn this number into a sample from the g-value distribution given by $F_g$.

More specifically, during generation, the LLM produces token $x_t$ by sampling from a modified distribution that incorporates the watermark:

$$x_t \sim P_t^\star(x \mid x_{<t}, \{g_{t,\ell}\}_\ell),$$

where $P_t^\star$ denotes the adapted token distribution. This modification preserves fluency while embedding structured signals useful for detection.

Given these constructions, the overall watermarking scheme is fully described by the tuple $(\mathcal{G}, r, P^\star)$. To ground this process in a hypothesis-testing framework, we now introduce the key assumption required for formal detectability analysis.

**Working Hypothesis 2.1 (Soundness of pseudorandomness in SynthID-Text).** In the watermarked LLM, the pseudorandom variables $g_{t,\ell}$ constructed above are i.i.d. across timesteps $t$, and are sampled from a base distribution (e.g., Bernoulli or Uniform). Furthermore, under the null hypothesis $H_0$, the variables $g_{t,\ell}$ are statistically independent of both the past context $x_{<t}$ and the generated token $x_t$.

## B   PREREQUISITES

**Theorem 13** (Lyapunov's Central Limit Theorem). *Let $X_1, X_2, \ldots, X_n$ be independent random variables with finite means $\mu_{t,\ell} = \mathbb{E}[X_i]$ and variances $\sigma_i^2 = \text{Var}(X_i)$. Define:*

$$S_n = \sum_{i=1}^n X_i, \quad \text{and} \quad B_n^2 = \sum_{i=1}^n \sigma_i^2$$

*If there exists $\delta > 0$ such that:*

$$\lim_{n \to \infty} \frac{1}{B_n^{2+\delta}} \sum_{i=1}^n \mathbb{E}\big[|X_i - \mu_{t,\ell}|^{2+\delta}\big] = 0$$

*then the normalized sum converges in distribution to a standard normal distribution:*

$$\frac{S_n - \sum_{i=1}^n \mu_{t,\ell}}{B_n} \xrightarrow{d} \mathcal{N}(0, 1)$$

**Theorem 14** (Theorem 32 in Dathathri et al. (2024a)). *The expected collision probability of a layer is no smaller than the collision probability of the previous layer for all tokens:*

$$\mathbb{E}[C_{\ell+1,t}] \geq C_{t,\ell},$$

*with equality hold if and only if $C_{t,\ell} = 1$.*

**Definition 3** (False Positive Rate (FPR) (Van Trees, 2004)). *Given the probability distribution of the unwatermarked distribution (Null Hypothesis) $p_{\neg w}(x)$, the FPR no smaller than $\tau$ is defined as:*

$$\mathbb{E}[FPR(\tau)] = p_{\neg w}(x \geq \tau) = 1 - CDF_{\neg w}(\tau). \tag{22}$$

*where $CDF_{\neg w}(\tau)$ is the CDF of the unwatermarked text.*

**Theorem 15** (Watermarked $g$-value distribution under Bernoulli(0.5) $g$-value distribution for Bayesian Score (Dathathri et al., 2024a)). *If $f_g = Bernoulli(0.5)$, then:*

$$f_{gw}(g_{t,\ell}) = f_g(g_{t,\ell}) \left( \hat{C}_{t,\ell} + (0.5 + g_{t,\ell})(1 - \hat{C}_{t,\ell}) \right) = \frac{1}{2}(\hat{C}_{t,\ell} + (0.5 + g_{t,\ell})(1 - \hat{C}_{t,\ell})).$$

**Theorem 16** (Watermarked $g$-value distribution under Uniform(0,1) $g$-value distribution for Bayesian Score (Dathathri et al., 2024a)). *If $f_g = Uniform[0, 1]$, then:*

$$f_{gw}(g_{t,\ell}) = f_g(g_{t,\ell}) \left( \hat{C}_{t,\ell} + 2g_{t,\ell}(1 - \hat{C}_{t,\ell}) \right) = \hat{C}_{t,\ell} + 2g_{t,\ell}(1 - \hat{C}_{t,\ell})$$

**Theorem 17** (Corollary 27 in Dathathri et al. (2024a)). *If $g$-value distribution $f_g$ is Bernoulli(p) for some $0 < p < 1$, then the watermarked $g$-value distribution given by the PDF is:*

$$f_{gw}(1) = p + p(1 - p)(1 - C_{t,\ell}).$$

*Hence, the watermarked g-value distribution is $Bernoulli\big(p + p(1 - p)(1 - C_{t,\ell})\big)$.*

## C  PROOFS FOR MEAN SCORE

### C.1  PROOF OF EQUATION (10) VIA CENTRAL LIMIT THEOREM

We prove that the Mean Score in Equation 1 ($\text{MS}(x) = \frac{1}{Tm} \sum_{t=1}^{T} \sum_{\ell=1}^{m} g_{t,\ell}$) converges in distribution to a normal distribution under the Central Limit Theorem (CLT) framework. We prove for the Bernoulli(0.5) distribution, as the proof for Uniform(0,1) distribution is identical.

We mainly use the Lyapunov's CLT in Theorem 13. In our setting, let $n = mT$, $X_i = g_{t,\ell} \sim$ Bernoulli($p_{t,\ell}$) are independent variables; $\mu_{t,\ell} = \mathbb{E}[g_{t,\ell}] = p_{t,\ell}$, $\sigma_{t,\ell}^2 = \text{var}(X_i) = \text{var}(g_{t,\ell}) = p_{t,\ell}(1 - p_{t,\ell})$ on watermarked data, where $p_{t,\ell} = \frac{3 - C_{t,\ell}}{4}$ (Theorem 3) and $C_{t,\ell}$ is a non-increasing function w.r.t. $\ell$ and its range is [0,1]. Now, we show that when $\delta = 2$,

$$\lim_{n \to \infty} \frac{1}{B_n^4} \left[ \sum_{t,\ell} \mathbb{E}[|g_{t,\ell} - \mu_{t,\ell}|^4] \right] = 0$$

**Step 1: Bounding the fourth central moment.** For $g_{t,\ell} \sim$ Bernoulli($p_{t,\ell}$), the fourth central moment is given by:

$$\mathbb{E}[(g_{t,\ell} - p_{t,\ell})^4] = p_{t,\ell}(1 - p_{t,\ell})(1 - 3p_{t,\ell} + 3p_{t,\ell}^2)$$

This function is maximized at $p_{t,\ell} = 0.5$, which yields:

$$\mathbb{E}[(g_{t,\ell} - p_{t,\ell})^4] \leq \frac{1}{16} \quad \forall \ell, t$$

Therefore, the sum over all $n$ terms is bounded as:

$$\sum_{i=1}^{n} \mathbb{E}[(g_{t,\ell} - \mu_{t,\ell})^4] \leq \frac{n}{16}$$

**Step 2: Bounding the total variance.** Since $p_{t,\ell} \in \left[\frac{1}{2}, \frac{3}{4}\right]$ (as $C_{t,\ell} \in [0, 1]$), we have:

$$\sigma_{t,\ell}^2 = p_{t,\ell}(1 - p_{t,\ell}) \in \left[\frac{3}{16}, \frac{1}{4}\right]$$

Thus, the total variance satisfies:

$$B_n^2 = \sum_{i=1}^n \sigma_{t,\ell}^2 \geq \frac{3n}{16} \quad \Rightarrow \quad B_n^4 \geq \left(\frac{3n}{16}\right)^2 = \frac{9n^2}{256}$$

**Step 3: Verifying the Lyapunov condition.** We compute the Lyapunov ratio:

$$\frac{1}{B_n^4} \sum_{i=1}^n \mathbb{E}[(g_{t,\ell} - \mu_{t,\ell})^4] \leq \frac{n \cdot \frac{1}{16}}{\frac{9n^2}{256}} = \frac{16}{9n} \to 0 \quad \text{as } n \to \infty$$

As the Lyapunov condition holds, and by the Lyapunov Central Limit Theorem, we have:

$$\frac{1}{\sqrt{n}} \sum_{i=1}^n \frac{g_{t,\ell} - \mu_{t,\ell}}{\sigma_{t,\ell}} \xrightarrow{d} \mathcal{N}(0,1) \quad \Rightarrow \quad \text{MS}(x) \xrightarrow{d} \mathcal{N}\left(\frac{1}{n}\sum_i \mu_{t,\ell}, \frac{1}{n^2}\sum_{i=1}^n \sigma_{t,\ell}^2\right)$$

Therefore, the Mean Score converges in distribution to a normal distribution as $n = Tm \to \infty$.

## C.2 PROOF OF PROPOSITION 1

From Equation 10, the PDF of the Mean Score follows a normal distribution, characterized by its corresponding mean and variance. Consequently, the CDF of the Mean Score is equivalent to that of a standard normal distribution after appropriate normalization.

To compute the expected TPR at a fixed FPR, i.e., FPR $= \epsilon$, we must consider the detection threshold $\tau(\epsilon)$ that achieves this FPR. Using this threshold and by Definition 1, we have

$$\mathbb{E}[\text{TPR}(\tau(\epsilon))|\text{FPR} = \epsilon] = \mathbb{P}_w(x \geq \tau(\epsilon)) = 1 - \text{CDF}_w(\tau(\epsilon)) = 1 - \Phi\left(\frac{\tau(\epsilon) - \mathbb{E}[\text{MS}(x)|w]}{\sqrt{\text{Var}[\text{MS}(x)|w]}}\right).$$

## C.3 PROOF OF THEOREM 3

We aim to compute the expected value and variance of the Mean Score under the watermarked distribution. Since the watermarked $g$-values, denoted as $g_{t,\ell}|w$, are independent random variables, the expected value of the Mean Score is given by:

$$\mathbb{E}\left[\text{MS}(x)|w\right] = \mathbb{E}\left[\frac{1}{mT}\sum_{t=1}^T \sum_{\ell=1}^m g_{t,\ell}(x)|w\right] = \frac{1}{mT}\sum_{t=1}^T \sum_{\ell=1}^m \mathbb{E}\left[g_{t,\ell}(x)|w\right].$$

The expectation of each watermarked $g$-value, $g_{t,\ell}|w$ can be expressed in terms of its probability mass function (PMF):

$$\mathbb{E}\left[g_{t,\ell}|w\right] = \sum_{g_{t,\ell} \in \{0,1\}} g_{t,\ell} \cdot f_{gw}(g_{t,\ell}) = f_{gw}(1),$$

First, we have $f_g(1) = 1/2, F_g(1) = 1$ for a Bernoulli(0,5) $g$-value distribution $f_g$, Based on the distribution given in Equation 7 of Theorem 1, and Bernoulli(0,5) $g$-value distribution $f_g$, we have:

$$f_{gw}(1) = f_g(1)\left[C_{t,\ell} + (1 - C_{t,\ell})(2F_g(1) - f_g(1)]\right] = \frac{1}{2}[C_{t,\ell} + \frac{3}{2}(1 - C_{t,\ell})] = \frac{3 - C_{t,\ell}}{4}.$$

Substituting this into the expectation expression concludes:

$$\mathbb{E}[\text{MS}(x)|w] = \frac{1}{mT}\sum_{t=1}^T \sum_{\ell=1}^m \frac{3 - C_{t,\ell}}{4}.$$

For convenience, we define $p_{t,\ell} = \frac{3 - C_{t,\ell}}{4}$, so that the expected Mean Score becomes:

$$\mathbb{E}[\text{MS}(x)|w] = \frac{1}{mT}\sum_{t=1}^T \sum_{\ell=1}^m p_{t,\ell}.$$

Since each $g_{t,\ell}|w \sim \text{Bernoulli}(p_{t,\ell})$, the variance of each is:

$$\text{Var}[g_{t,\ell}|w] = p_{t,\ell}(1 - p_{t,\ell}).$$

Using the linearity of variance for independent variables, the variance of the Mean Score is:

$$\text{Var}[\text{MS}(x)|w] = \text{Var}\left[\frac{1}{mT}\sum_{t=1}^{T}\sum_{\ell=1}^{m}g_{t,\ell}|w\right] = \left(\frac{1}{mT}\right)^2\sum_{t=1}^{T}\sum_{\ell=1}^{m}\text{Var}(g_{t,\ell}|w) = \left(\frac{1}{mT}\right)^2\sum_{t=1}^{T}\sum_{\ell=1}^{m}p_{t,\ell}(1 - p_{t,\ell}).$$

### C.4 PROOF OF THEOREM 4

Let $g_{t,\ell}(x) \sim \text{Uniform}(0, 1)$. We have its PDF $f_g(x) = 1$ and CDF $F_g(x) = x$.

The expected value of watermarked Mean Score is given by:

$$\mathbb{E}[\text{MS}(x)|w] = \mathbb{E}\left[\frac{1}{mT}\sum_{t=1}^{T}\sum_{\ell=1}^{m}g_{t,\ell}(x)|w\right] = \frac{1}{mT}\sum_{t,\ell}\mathbb{E}[g_{t,\ell}|w]$$

We show the expected value of $g_{t,\ell}|w$ below:

$$\mathbb{E}[g_{t,\ell}|w] = \int_{0}^{1} x f_{gw}(x)\, dx$$

$$\stackrel{\text{Eqn6}}{=} \int_{0}^{1} x f_g(x)\left[C_{t,\ell} + 2(1 - C_{t,\ell})F_g(x)\right] dx$$

$$= \int_{0}^{1} x\left[C_{t,\ell} + 2(1 - C_{t,\ell})x\right] dx$$

$$= C_{t,\ell}\int_{0}^{1} x\,dx + 2(1 - C_{t,\ell})\int_{0}^{1} x^2\,dx$$

$$= \frac{1}{2}C_{t,\ell} + \frac{1}{3}\cdot 2(1 - C_{t,\ell})$$

$$= \frac{4 - C_{t,\ell}}{6}$$

Hence, $\mathbb{E}[\text{MS}(x)|w] = \frac{1}{mT}\sum_{t,\ell}\mathbb{E}[g_{t,\ell}|w] = \frac{1}{mT}\sum_{t,\ell}\frac{4 - C_{t,\ell}}{6}$.

For convenience, we define $p_{t,\ell} = \frac{4 - C_{t,\ell}}{6}$, so that the expected Mean Score becomes:

$$\mathbb{E}[\text{MS}(x)|w] = \frac{1}{mT}\sum_{t,\ell}p_{t,\ell}.$$

For the the variance, under $\text{Uniform}(0, 1)$ watermark scores $g_{t,\ell} \sim \text{Uniform}(0, 1)$, note that the variance of $g_{t,\ell}$ is $\text{Var}[g_{t,\ell}|w] = \mathbb{E}[(g_{t,\ell}|w)^2] - (\mathbb{E}[g_{t,\ell}|w])^2$.

Given $g_{t,\ell}|w \sim f_{gw}(x)$, we have

$$\mathbb{E}[(g_{t,\ell}|w)^2] = \int_{0}^{1} x^2 f_{gw}(x)\, dx$$

$$= \int_{0}^{1} x^2 f_g(x)\left[C_{t,\ell} + 2(1 - C_{t,\ell})F_g(x)\right] dx$$

$$= \int_{0}^{1} x^2\left[C_{t,\ell} + 2(1 - C_{t,\ell})x\right] dx$$

$$= C_{t,\ell}\int_{0}^{1} x^2\,dx + 2(1 - C_{t,\ell})\int_{0}^{1} x^3\,dx$$

$$= \frac{1}{3}C_{t,\ell} + \frac{1}{4}\cdot 2(1 - C_{t,\ell})$$

$$= \frac{3 - C_{t,\ell}}{6} = p_{t,\ell} - \frac{1}{6}$$

Therefore,

$$\text{Var}[\text{MS}(x)|w] = \left(\frac{1}{mT}\right)^2 \sum_{t,\ell} \left[p_{t,\ell}(1 - p_{t,\ell}) - \frac{1}{6}\right].$$

## C.5 PROOF OF THEOREM 5

Under the null hypothesis $H_0$ (non-watermarked text $\neg w$), each component $g_{t,\ell}$ of the Mean Score is drawn from a Bernoulli$(0.5)$ distribution. The mean and variance of each $g_{t,\ell}$ are respectively:

$$\mathbb{E}[g_{t,\ell}] = \frac{1}{2}, \quad \text{Var}(g_{t,\ell}) = \frac{1}{4}.$$

Recall the distribution of Mean Score is a Normal distribution under central limit theorem. For unwatermarked texts, we have the probability distribution as:

$$\text{MS}(x)|\neg w \sim \mathcal{N}\left(\frac{1}{2}, \frac{1}{4mT}\right).$$

We aim to find the threshold $\tau$ such that the probability of a false positive is equal to a given level $\epsilon$.

Based on Definition 3, we can estimate the threshold knowing the expected value of the FPR:

$$\mathbb{E}[\text{FPR}(\tau(\epsilon)) = \epsilon] = \mathbb{P}_{\neg w}(x \geq \tau(\epsilon)) = 1 - \text{CDF}_{\neg w}(\tau(\epsilon)) = 1 - \Phi\left(\frac{\tau(\epsilon) - \mathbb{E}[\text{MS}(x)|\neg w]}{\sqrt{\text{Var}[\text{MS}(x)|\neg w]}}\right)$$

$$\iff \mathbb{P}(\text{MS}(x) \geq \tau(\epsilon)|x \sim H_0) = \epsilon.$$

Standardizing:

$$\mathbb{P}\left(\frac{\text{MS}(x)|\neg w - \frac{1}{2}}{\sqrt{1/(4mT)}} > \frac{\tau(\epsilon) - \frac{1}{2}}{\sqrt{1/(4mT)}}\right) = \epsilon,$$

which implies:

$$\Phi\left(\frac{\tau(\epsilon) - \frac{1}{2}}{1/2\sqrt{mT}}\right) = 1 - \epsilon.$$

Hence, solving for $\tau$ gives:

$$\epsilon = 1 - \Phi\left(\frac{\tau(\epsilon) - \frac{1}{2}}{1/2\sqrt{mT}}\right) \Rightarrow \frac{\tau(\epsilon) - \frac{1}{2}}{1/2\sqrt{mT}} = \Phi^{-1}(1 - \epsilon) \implies \tau(\epsilon) = \frac{1}{2} + \frac{\Phi^{-1}(1 - \epsilon)}{2\sqrt{mT}}.$$

where $\Phi$ denotes the standard normal CDF.

## C.6 PROOF OF THEOREM 6

Under the null hypothesis $H_0$ (non-watermarked text $\neg w$), each component $g_{t,\ell}$ of the Mean Score is drawn from a Uniform$(0, 1)$ distribution. The mean and variance of each $g_{t,\ell}$ are respectively:

$$\mathbb{E}[g_{t,\ell}] = \frac{1}{2}, \quad \text{Var}(g_{t,\ell}) = \frac{1}{12}.$$

Since the Mean Score is computed over $mT$ i.i.d. components, by the Central Limit Theorem, the sample mean converges in distribution to a normal distribution:

$$\text{MS}(x)|\neg w \sim \mathcal{N}\left(\frac{1}{2}, \frac{1}{12mT}\right).$$

We want to set a threshold $\tau$ such that the false positive rate (FPR) is equal to $\epsilon$:

$$\mathbb{E}[\text{FPR}(\tau(\epsilon)) = \epsilon] = 1 - \Phi\left(\frac{\tau(\epsilon) - \mathbb{E}[\text{MS}(x)|\neg w]}{\sqrt{\text{Var}[\text{MS}(x)|\neg w]}}\right) \iff \mathbb{P}(\text{MS}(x) \geq \tau(\epsilon)|x \sim H_0) = \epsilon.$$

Standardizing:

$$\mathbb{P}\left(\frac{\mathrm{MS}(x)|\neg w - \frac{1}{2}}{\sqrt{1/(12mT)}} > \frac{\tau(\epsilon) - \frac{1}{2}}{\sqrt{1/(12mT)}}\right) = \epsilon,$$

which implies:

$$\Phi\left(\frac{\tau(\epsilon) - \frac{1}{2}}{1/\sqrt{12mT}}\right) = 1 - \epsilon.$$

Solving for $\tau$, we get:

$$\tau(\epsilon) = \frac{1}{2} + \frac{\Phi^{-1}(1-\epsilon)}{\sqrt{12mT}}.$$

## C.7 Proof of Theorem 7

**Bernoulli(0.5):** Assume the $g$-values are sampled from a Bernoulli(0.5) distribution and let $a_{t,\ell} = 1 - C_{t,\ell}$. Replacing it into the Equation in Theorem 3, we have

$$\mathbb{E}[\mathrm{MS}(x)|w] = \frac{1}{2} + \frac{1}{4mT}\sum_{t,\ell} a_{t,\ell} \quad \text{and} \quad \mathrm{Var}[\mathrm{MS}(x)|w] = \frac{1}{(2mT)^2}\sum_{t,\ell}(4 - a_{t,\ell}^2)$$

By the CLT, the expected TPR under FPR $= \epsilon (\in [0.01, 0.5)$ is approximated by:

$$\mathbb{E}[\mathrm{TPR}(\tau(\epsilon))|\mathrm{FPR} = \epsilon] = 1 - \Phi\left(\frac{\tau(\epsilon) - \mathbb{E}[\mathrm{MS}(x)|w]}{\sqrt{\mathrm{Var}[\mathrm{MS}(x)|w]}}\right)$$

Substituting the threshold $\tau(\epsilon) = \frac{1}{2} + \frac{\Phi^{-1}(1-\epsilon)}{2\sqrt{mT}}$ from Theorem 5, we have

$$\mathbb{E}[\mathrm{TPR}(\tau(\epsilon))|\mathrm{FPR} = \epsilon] = 1 - \Phi\left(\frac{\frac{1}{2} + \frac{\Phi^{-1}(1-\epsilon)}{2\sqrt{mT}} - \frac{1}{2} - \frac{1}{4mT}\sum_{t,\ell} a_{t,\ell}}{\sqrt{\frac{1}{(2mT)^2}\sum_{t,\ell}(4 - a_{t,\ell}^2)}}\right)$$

$$= 1 - \Phi\left(\frac{2\Phi^{-1}(1-\epsilon)\sqrt{mT} - \sum_{t,\ell} a_{t,\ell}}{2\sqrt{4mT - \sum_{t,\ell} a_{t,\ell}^2}}\right)$$

$$= 1 - \Phi\left(\frac{2\Phi^{-1}(1-\epsilon)\sqrt{mT} - mT \cdot \mathbb{E}[a_{t,\ell}]}{2\sqrt{4mT - mT \cdot \mathbb{E}[a_{t,\ell}^2]}}\right)$$

Where $\mathbb{E}[a_{t,\ell}]$ and $\mathbb{E}[a_{t,\ell}^2]$ are approximately constant. Letting:

$$\hat{A} = 2\Phi^{-1}(1-\epsilon)\sqrt{T}, \quad A = T \cdot \mathbb{E}[a_{t,\ell}]. \quad B = 2\sqrt{T \cdot (4 - \mathbb{E}[a_{t,\ell}^2])},$$

Then:

$$\mathbb{E}[\mathrm{TPR}(\tau(\epsilon))|\mathrm{FPR} = \epsilon] = 1 - \Phi\left(\frac{-Am + \hat{A}\sqrt{m}}{B\sqrt{m}}\right)$$

Moreover, if, for the first time $M$ such that $C_{M,t} = 1$, then $a_{M,t} = 0$ for all $t$. Note that $C_{t,\ell}$ is a non-decreasing function w.r.t. the layer number $l$ (see Theorem 14), hence $C_{t,\ell} = 1$ (and $a_{t,\ell} = 0$) for all $l \geq M$. This implies $\sum_{t,\ell} a_{t,\ell}$ and $\sum_{t,\ell} a_{t,\ell}^2$ become constant, due to:

$$\text{for m>M: } \sum_{t,\ell} a_{t,\ell} = \sum_{t,\ell}(1 - C_{t,\ell}) = \sum_t \sum_{\ell=1}^{\ell=M}(1 - C_{t,\ell}) + 0 = \sum_t \sum_{\ell=1}^{\ell=M}(1 - C_{t,\ell}) \text{ is constant}$$

$$\text{for m>M : } \sum_{t,\ell} a_{t,\ell}^2 = \sum_{t,\ell}(1 - C_{t,\ell})^2 = \sum_t \sum_{\ell=1}^{\ell=M}(1 - C_{\ell=1,t})^2 + 0 = \sum_t \sum_{\ell=1}^{\ell=M}(1 - C_{t,\ell})^2 \text{ is constant}$$

In this case, the expression simplifies as follows:

$$\mathbb{E}[\text{TPR}(\tau(\epsilon))|\text{FPR} = \epsilon] = 1 - \Phi\left(\frac{\hat{A}\sqrt{m} - \hat{B}}{2\sqrt{\hat{C}m - \hat{D}}}\right)$$

Where: $\hat{B} = \sum_{t,\ell} a_{t,\ell}, \hat{C} = 4T, \hat{D} = \sum_{t,\ell} a_{t,\ell}^2$.

**Uniform[0,1]:** Assume the $g$-values are sampled from a Uniform[0,1] distribution and let $a_{t,\ell} = 1 - C_{t,\ell}$. Similarly, based on Theorem 4 the expected mean score and variance become:

$$\mathbb{E}[\text{MS}(x)|w] = \frac{1}{2} + \frac{1}{6mT}\sum_{t,\ell} a_{t,\ell} \quad \text{and} \quad \text{Var}[\text{MS}(x)|w] = \frac{1}{(6mT)^2}\sum_{t,\ell}(3 - a_{t,\ell}^2)$$

Hence, given threshold $\tau(\epsilon) = \frac{1}{2} + \frac{\Phi^{-1}(1-\epsilon)}{\sqrt{12mT}}$ in Theorem 6 we have:

$$\mathbb{E}[\text{TPR}(\tau(\epsilon))|\text{FPR} = \epsilon] = 1 - \Phi\left(\frac{\sqrt{3}\Phi^{-1}(1-\epsilon)\sqrt{mT} - mT \cdot \mathbb{E}[a_{t,\ell}]}{\sqrt{3mT - mT \cdot \mathbb{E}[a_{t,\ell}^2]}}\right)$$

Where $\mathbb{E}[a_{t,\ell}]$ and $\mathbb{E}[a_{t,\ell}^2]$ are approximately constant. Letting:

$$\hat{A} = \sqrt{3}\Phi^{-1}(1-\epsilon)\sqrt{T}, \quad A = T \cdot \mathbb{E}[a_{t,\ell}]. \quad B = \sqrt{T \cdot (3 - \mathbb{E}[a_{t,\ell}^2])},$$

Then:

$$\mathbb{E}[\text{TPR}(\tau(\epsilon))|\text{FPR} = \epsilon] = 1 - \Phi\left(\frac{-Am + \hat{A}\sqrt{m}}{B\sqrt{m}}\right)$$

Moreover, if for the first time $M$ such that $C_{M,t} = 1$, then $a_{M,t} = 0$ for all $t$, $\sum_{t,\ell} a_{t,\ell}$ and $\sum_{t,\ell} a_{t,\ell}^2$ become constant. In this case, the expression simplifies as follows:

$$\mathbb{E}[\text{TPR}(\tau(\epsilon))|\text{FPR} = \epsilon] = 1 - \Phi\left(\frac{\hat{A}\sqrt{m} - \hat{B}}{\sqrt{\hat{C}m - \hat{D}}}\right)$$

Where: $\hat{B} = \sum_{t,\ell} a_{t,\ell}, \hat{C} = 3T, \hat{D} = \sum_{t,\ell} a_{t,\ell}^2$.

As a conclusion, we have:

$$\mathbb{E}\left[\text{TPR}(\tau(\epsilon))|\text{FPR} = \epsilon\right] = \begin{cases} 1 - \Phi\left(\dfrac{-Am + \hat{A}\sqrt{m}}{B\sqrt{m}}\right) & \text{if } m < M \\ 1 - \Phi\left(\dfrac{\hat{A}\sqrt{m} - \hat{B}}{\sqrt{\hat{C}m - \hat{D}}}\right) & \text{if } m \geq M \end{cases}$$

## C.8  PROOF OF COROLLARY 1

To prove that the expected TPR is a unimodal function w.r.t. $m$, we calculate the derivative of TPR.

$$\text{For } m < M, \quad \frac{d}{dm}\left[\frac{-Am + \hat{A}\sqrt{m}}{B\sqrt{m}}\right] = \frac{(-A + \frac{\hat{A}}{2\sqrt{m}})B\sqrt{m} - (-Am + \hat{A}\sqrt{m})\frac{B}{2\sqrt{m}}}{B^2 m} = \frac{-A}{2B\sqrt{m}} < 0$$

$$\text{For } m \geq M, \quad \frac{d}{dm}\left[\frac{\hat{A}\sqrt{m} - \hat{B}}{\sqrt{\hat{C}m - \hat{D}}}\right] = \frac{\frac{\hat{A}}{2\sqrt{m}}\sqrt{\hat{C}m - \hat{D}} - \left(\hat{A}\sqrt{m} - \hat{B}\right)\frac{\hat{C}}{2\sqrt{\hat{C}m - \hat{D}}}}{\hat{C}m - \hat{D}}$$

$$= \frac{\hat{B}\hat{C} - \hat{A}\hat{D}}{\hat{C}m - \hat{D}} \propto \frac{T\sum_{t,\ell} a_{t,\ell} - \sqrt{T}\sum_{t,\ell} a_{t,\ell}^2}{mT - \sum_{t,\ell} a_{t,\ell}^2} > 0$$

This means the term inside $\Phi$ function first decreases and then increases, implying the expected TPR is a unimodal function (under both Bernoulli(0.5) and Uniform(0,1)). Further, due to behavior of $\Phi$ and the negative sign, we conclude that TPR first increases and then decreases.

## C.9 PROOF OF COROLLARY 2

Since the expected TPR first increases then decreases, the max value happens at where the trend of TPR changes which happens at $m = M$ where $C_{M,t}$ becomes 1 for the first time for all $t$.

For the final TPR value, we set $m$ to be infinity:

$$\lim_{m \to \infty} \mathbb{E}[\text{TPR}(\tau(\epsilon))|\text{FPR} = \epsilon] = \lim_{m \to \infty} 1 - \Phi\left(\frac{\hat{A}\sqrt{m} - \hat{B}}{\sqrt{\hat{C}m - \hat{D}}}\right)$$

$$= 1 - \Phi\left(\lim_{m \to \infty} \frac{\hat{A}\sqrt{m} - \hat{B}}{\sqrt{\hat{C}m - \hat{D}}}\right)$$

$$= 1 - \Phi\left(\frac{\hat{A}}{\sqrt{\hat{C}}}\right)$$

$$= 1 - \Phi\left(\Phi^{-1}(1 - \epsilon)\right) = \epsilon = \text{FPR}$$

# D PROOFS FOR BAYESIAN SCORE

## D.1 PROOF OF EQUATION (14) VIA CENTRAL LIMIT THEOREM

Based on Equation 3, and let $X = \frac{P(g|w)}{p(g|\neg w)}$ and $\alpha = \frac{P(w)}{P(\neg w)}$ we can simplify Bayesian score as:

$$\text{BS}(x) = \sigma[\log(\alpha X)] = \frac{1}{1 + e^{-\log(\alpha X)}} = \frac{\alpha X}{\alpha X + 1}$$

Then via inverting the transformation, we have:

$$CDF_{BS}(x) = CDF_X\left(\frac{x}{\alpha(1-x)}\right)$$

Now we need to find CDF of $X$. Lets first define $f_{gw}$ as follow:

$$f_{gw}(g_{t,\ell}) = \sum_{C_{t,\ell}=1}^{2} P(g_{t,\ell}|\psi_{t,\ell} = C_{t,\ell})P(\psi_{t,\ell} = C_{t,\ell}|g_{t,<\ell})$$

By using Equation 8, we have:

$$X = \frac{P(g|w)}{p(g|\neg w)} = \prod_{t,\ell} \frac{f_{gw}(g_{t,\ell})}{f_g(g_{t,\ell})} \Rightarrow \log X = \sum_{t,\ell} \log\left(\frac{f_{gw}(g_{t,\ell})}{f_g(g_{t,\ell})}\right) \Rightarrow X = e^{\sum_{t,\ell} \log\left(\frac{f_{gw}(g_{t,\ell})}{f_g(g_{t,\ell})}\right)}$$

If we consider each term $\log\left(\frac{f_{gw}(g_{t,\ell})}{f_g(g_{t,\ell})}\right)$ as a random variable $X_{t,\ell}$, then $X$ is approximately a log-normal distribution based on central limit theorem. That is,

$$X \sim \text{LogNormal}\left(\mathbb{E}[\sum_{t,\ell} \log(X_{t,\ell})], \text{Var}(\sum_{t,\ell} \log(X_{t,\ell}))\right).$$

## D.2 PROOF OF PROPOSITION 2

From Definition 1 we know that:

$$\mathbb{E}[\text{TPR}(\tau(\epsilon))] = p_w(x \geq \tau(\epsilon)) = 1 - CDF_{BS|w}(\tau(\epsilon)),$$

Based on C.1, we have

$$CDF_X(x) = CDF_{LogNormal}(x) = \Phi\left(\frac{\ln(x) - \mathbb{E}[\sum_{t,\ell} \log(X_{t,\ell})]}{\sqrt{\text{Var}[\sum_{t,\ell} \log(X_{t,\ell})]}}\right)$$

$$= \Phi\left(\frac{\ln(x) - \mathbb{E}\left[\log\left(\frac{P(g|\omega)}{P(g|\neg\omega)}\right)\right]}{\sqrt{\text{Var}\left[\log\left(\frac{P(g|\omega)}{P(g|\neg\omega)}\right)\right]}}\right)$$

$$= \Phi\left(\frac{\ln(x) - \mathbb{E}[\text{BS}(x)]}{\sqrt{\text{Var}[\text{BS}(x)]}}\right)$$

Therefore, the CDF of Bayesian Score based on the Central Limit Theorem is given by:

$$CDF_{BS|w}(x) = CDF_{X|w}\left(\frac{x}{\alpha(1-x)}\right) = \Phi\left(\frac{\ln\left(\frac{x}{\alpha(1-x)}\right) - \mathbb{E}[BS(x)|w]}{\sqrt{\mathrm{Var}[BS(x)|w]}}\right),$$

### D.3 Proof of Theorem 8

**Watermarked Data:** Based on Theorem 15, we can expand its definition of Bayesian Score on watermarked data as:

$$\mathbb{E}[\mathrm{BS}(x)|w] = \mathbb{E}\left[\log\left(\frac{P(g|w)}{P(g|\neg w)}\right)\right] = \mathbb{E}\left[\sum_{t,\ell}\log\left(\frac{f_{gw}(g_{t,\ell})}{f_g(g_{t,\ell})}\right)\right]$$

$$= \sum_{t,\ell}\mathbb{E}\left[\log\left(\hat{C}_{t,\ell} + (0.5 + g_{t,\ell})(1 - \hat{C}_{t,\ell})\right)\right],$$

where $g_{t,\ell} \sim f_{gw}$. Then,

$$\mathbb{E}\left[\log\left(\hat{C}_{t,\ell} + (0.5 + g_{t,\ell})(1 - \hat{C}_{t,\ell})\right)\right] = \sum_{g_{t,\ell}=0,1} f_{gw}(g_{t,\ell})\log\left(\hat{C}_{t,\ell} + (0.5 + g_{t,\ell})(1 - \hat{C}_{t,\ell})\right)$$

$$= f_{gw}(0)\log\left(\frac{1 + \hat{C}_{t,\ell}}{2}\right) + f_{gw}(1)\log\left(\frac{3 - \hat{C}_{t,\ell}}{2}\right)$$

$$= \frac{1 + C_{t,\ell}}{4}\log\left(\frac{1 + \hat{C}_{t,\ell}}{2}\right) + \frac{3 - C_{t,\ell}}{4}\log\left(\frac{3 - \hat{C}_{t,\ell}}{2}\right)$$

Therefore:

$$\mathbb{E}[\mathrm{BS}(x)|w] = \sum_{t,\ell}\frac{1 + C_{t,\ell}}{4}\log\left(\frac{1 + \hat{C}_{t,\ell}}{2}\right) + \frac{3 - C_{t,\ell}}{4}\log\left(\frac{3 - \hat{C}_{t,\ell}}{2}\right)$$

For the variance, note that $g$-values are independent, we have:

$$\mathrm{Var}[\mathrm{BS}(x)|w] = \mathrm{Var}\left[\log\left(\frac{p(g|w)}{p(g|\neg w)}\right)\right] = \mathrm{Var}\left[\sum_{t,\ell}\log\left(\frac{f_{gw}(g_{t,\ell})}{f_g(g_{t,\ell})}\right)\right]$$

$$= \sum_{t,\ell}\mathrm{Var}\left[\log\left(\hat{C}_{t,\ell} + (0.5 + g_{t,\ell})(1 - \hat{C}_{t,\ell})\right)\right]$$

For each variance, by applying $\mathrm{Var}(X) = \mathbb{E}[X^2] - (\mathbb{E}[X])^2$, we have:

$$\mathrm{Var}\left[\log\left(\hat{C}_{t,\ell} + (0.5 + g_{t,\ell})(1 - \hat{C}_{t,\ell})\right)\right] = \sum_{g_{\ell.t}=0,1} f_{gw}(g_{t,\ell})\log^2\left(\hat{C}_{t,\ell} + (0.5 + g_{t,\ell})(1 - \hat{C}_{t,\ell})\right) - \mu^2$$

$$= f_{gw}(0)\log^2\left(\frac{1 + \hat{C}_{t,\ell}}{2}\right) + f_{gw}(1)\log^2\left(\frac{3 - \hat{C}_{t,\ell}}{2}\right) - \mu^2$$

$$= \frac{1 + C_{t,\ell}}{4}\log^2\left(\frac{1 + \hat{C}_{t,\ell}}{2}\right) + \frac{3 - C_{t,\ell}}{4}\log^2\left(\frac{3 - \hat{C}_{t,\ell}}{2}\right) - \mu^2$$

where $\mu = \mathbb{E}\left[\log\left(\hat{C}_{t,\ell} + (0.5 + g_{t,\ell})(1 - \hat{C}_{t,\ell})\right)\right]$. Hence, we have:

$$\mathrm{Var}[BS(x)|w] = \sum_{t,\ell}\left[\frac{3 - C_{t,\ell}}{4} \cdot \log^2\left(\frac{3 - \hat{C}_{t,\ell}}{2}\right) + \frac{1 + C_{t,\ell}}{4} \cdot \log^2\left(\frac{\hat{C}_{t,\ell} + 1}{2}\right)\right.$$

$$\left. - \left[\frac{3 - C_{t,\ell}}{4} \cdot \log\left(\frac{3 - \hat{C}_{t,\ell}}{2}\right) + \frac{1 + C_{t,\ell}}{4} \cdot \log\left(\frac{1 + \hat{C}_{t,\ell}}{2}\right)\right]^2\right]$$

**Unwatermarked Data:** Based on Theorem 15 we can expand the definition of Bayesian Score on unwatermarked data as:

$$\mathbb{E}[\text{BS}(x)|\neg w] = \sum_{t,\ell} \mathbb{E}\left[\log\left(\hat{C}_{t,\ell} + (0.5 + g_{t,\ell})(1 - \hat{C}_{t,\ell})\right)\right], \quad g_{t,\ell} \sim f_g.$$

Then,

$$\mathbb{E}\left[\log\left(\hat{C}_{t,\ell} + (0.5 + g_{t,\ell})(1 - \hat{C}_{t,\ell})\right)\right] = \sum_{g_{\ell,t}=0,1} f_g(g_{t,\ell})\log\left(\hat{C}_{t,\ell} + (0.5 + g_{t,\ell})(1 - \hat{C}_{t,\ell})\right)$$

$$= f_g(0)\log\left(\frac{1 + \hat{C}_{t,\ell}}{2}\right) + f_g(1)\log\left(\frac{3 - \hat{C}_{t,\ell}}{2}\right)$$

$$= \frac{1}{2}\log\left(\frac{1 + \hat{C}_{t,\ell}}{2}\right) + \frac{1}{2}\log\left(\frac{3 - \hat{C}_{t,\ell}}{2}\right)$$

Therefore, we have:

$$\mathbb{E}[\text{BS}(x)|\neg w] = \sum_{t,\ell} \frac{1}{2}\log\left(\frac{1 + \hat{C}_{t,\ell}}{2}\right) + \frac{1}{2}\log\left(\frac{3 - \hat{C}_{t,\ell}}{2}\right)$$

For the variance, as $g$-values are independent, we have:

$$\text{Var}[\text{BS}(x)|\neg w] = \sum_{t,\ell} \text{Var}\left[\log\left(\hat{C}_{t,\ell} + (0.5 + g_{t,\ell})(1 - \hat{C}_{t,\ell})\right)\right], \quad g_{t,\ell} \sim f_g.$$

For each variance we have:

$$\text{Var}\left[\log\left(\hat{C}_{t,\ell} + (0.5 + g_{t,\ell})(1 - \hat{C}_{t,\ell})\right)\right] = \sum_{g_{\ell,t}=0,1} f_g(g_{t,\ell})\log^2\left(\hat{C}_{t,\ell} + (0.5 + g_{t,\ell})(1 - \hat{C}_{t,\ell})\right) - \mu^2$$

$$= f_g(0)\log^2\left(\frac{1 + \hat{C}_{t,\ell}}{2}\right) + f_g(1)\log^2\left(\frac{3 - \hat{C}_{t,\ell}}{2}\right) - \mu^2$$

$$= \frac{1}{2}\log^2\left(\frac{1 + \hat{C}_{t,\ell}}{2}\right) + \frac{1}{2}\log^2\left(\frac{3 - \hat{C}_{t,\ell}}{2}\right) - \mu^2$$

where $\mu = \mathbb{E}[\log\left(\hat{C}_{t,\ell} + (0.5 + g_{t,\ell})(1 - \hat{C}_{t,\ell})\right)]$.

Hence we have:

$$\text{Var}[BS(x)|\neg w] = \sum_{t,\ell}\left[\frac{1}{2}\cdot\log^2\left(\frac{3 - \hat{C}_{t,\ell}}{2}\right) + \frac{1}{2}\cdot\log^2\left(\frac{\hat{C}_{t,\ell} + 1}{2}\right) - \left[\frac{1}{2}\cdot\log\left(\frac{3 - \hat{C}_{t,\ell}}{2}\right) + \frac{1}{2}\cdot\log\left(\frac{1 + \hat{C}_{t,\ell}}{2}\right)\right]^2\right]$$

### D.4 PROOF OF THEOREM 9

Defining $I_{1_{t,\ell}}, I_{2_{t,\ell}}, I_{3_{t,\ell}}, I_{4_{t,\ell}}$ as follow:

$$I_{1_{t,\ell}} = \int_0^1 \log(\hat{C}_{t,\ell} + 2(1 - \hat{C}_{t,\ell})x)dx, \quad I_{2_{t,\ell}} = \int_0^1 x\log(\hat{C}_{t,\ell} + 2(1 - \hat{C}_{t,\ell})x)dx$$

$$I_{3_{t,\ell}} = \int_0^1 \log^2(\hat{C}_{t,\ell} + 2(1 - \hat{C}_{t,\ell})x)dx, \quad I_{4_{t,\ell}} = \int_0^1 x\log^2(\hat{C}_{t,\ell} + 2(1 - \hat{C}_{t,\ell})x)dx$$

**Watermarked Data:** We can then calculate the expected value and variance of Bayesian Score for watermarked data. Similarly, based on Theorem 16 and $g_{t,\ell} \sim f_{gw}$, we have:

$$\mathbb{E}[\text{BS}(x)|w] = \mathbb{E}\left[\sum_{t,\ell}\log\left(\frac{f_{gw}(g_{t,\ell})}{f_g(g_{t,\ell})}\right)\right] = \sum_{t,\ell}\mathbb{E}\left[\log\left(\hat{C}_{t,\ell} + 2(1 - \hat{C}_{t,\ell})g_{t,\ell}\right)\right]$$

Hence, we have:

$$\mathbb{E}\left[\log\left(\hat{C}_{t,\ell} + 2(1-\hat{C}_{t,\ell})g_{t,\ell}\right)\right] = \int_0^1 f_{gw}(g_{t,\ell})\log\left(\hat{C}_{t,\ell} + 2(1-\hat{C}_{t,\ell})g_{t,\ell}\right) dg_{t,\ell}$$

$$= \int_0^1 \left[C_{t,\ell} + 2(1-C_{t,\ell})g_{t,\ell}\right](g_{t,\ell})\log\left(\hat{C}_{t,\ell} + 2(1-\hat{C}_{t,\ell})g_{t,\ell}\right) dg_{t,\ell}$$

$$= C_{t,\ell} \cdot I_{1_{t,\ell}} + 2(1-C_{t,\ell})I_{2_{t,\ell}}$$

and

$$\mathbb{E}[\mathrm{BS}(x)|w] = \sum_{t,\ell} C_{t,\ell} \cdot I_{1_{t,\ell}} + 2(1-C_{t,\ell})I_{2_{t,\ell}}$$

For the variance we have:

$$\mathrm{Var}[\mathrm{BS}(x)|w] = \mathrm{Var}\left[\sum_{t,\ell}\log\left(\frac{f_{gw}(g_{t,\ell})}{f_g(g_{t,\ell})}\right)\right] = \sum_{t,\ell}\mathrm{Var}\left[\log\left(\hat{C}_{t,\ell} + 2(1-\hat{C}_{t,\ell})g_{t,\ell}\right)\right]$$

where

$$\mathrm{Var}\left[\log\left(\hat{C}_{t,\ell} + 2(1-\hat{C}_{t,\ell})g_{t,\ell}\right)\right] = \int_0^1 f_{gw}(g_{t,\ell})\log^2\left(\hat{C}_{t,\ell} + 2(1-\hat{C}_{t,\ell})g_{t,\ell}\right) dg_{t,\ell} - \mu^2$$

$$= \int_0^1 \left[C_{t,\ell} + 2(1-C_{t,\ell})g_{t,\ell}\right]\log^2\left(\hat{C}_{t,\ell} + 2(1-\hat{C}_{t,\ell})g_{t,\ell}\right) dg_{t,\ell} - \mu^2$$

$$= C_{t,\ell} \cdot I_{3_{t,\ell}} + 2(1-C_{t,\ell})I_{4_{t,\ell}} - (C_{t,\ell} \cdot I_{1_{t,\ell}} + 2(1-C_{t,\ell})I_{2_{t,\ell}})^2$$

Therefore:

$$\mathrm{Var}[\mathrm{BS}(x)|w] = \sum_{t,\ell} C_{t,\ell} \cdot I_{3_{t,\ell}} + 2(1-C_{t,\ell})I_{4_{t,\ell}} - (C_{t,\ell} \cdot I_{1_{t,\ell}} + 2(1-C_{t,\ell})I_{2_{t,\ell}})^2$$

**Unwatermarked Data:** We calculate the expected value and variance of Bayesian Score for unwatermarked data. Based on Theorem 16 and $g_{t,\ell} \sim f_g$, we have:

$$\mathbb{E}[\mathrm{BS}(x)|\neg w] = \sum_{t,\ell}\mathbb{E}\left[\log\left(\hat{C}_{t,\ell} + 2(1-\hat{C}_{t,\ell})g_{t,\ell}\right)\right]$$

$$= \sum_{t,\ell}\int_0^1 f_{gw}(g_{t,\ell})\log\left(\hat{C}_{t,\ell} + 2(1-\hat{C}_{t,\ell})g_{t,\ell}\right) dg_{t,\ell}$$

$$= \sum_{t,\ell}\int_0^1 \log\left(\hat{C}_{t,\ell} + 2(1-\hat{C}_{t,\ell})g_{t,\ell}\right) dg_{t,\ell}$$

$$= \sum_{t,\ell} I_{1_{t,\ell}}$$

For the variance we have:

$$\mathrm{Var}[\mathrm{BS}(x)|\neg w] = \sum_{t,\ell}\mathrm{Var}\left[\log\left(\hat{C}_{t,\ell} + 2(1-\hat{C}_{t,\ell})g_{t,\ell}\right)\right]$$

$$= \sum_{t,\ell}\int_0^1 f_{gw}(g_{t,\ell})\log^2\left(\hat{C}_{t,\ell} + 2(1-\hat{C}_{t,\ell})g_{t,\ell}\right) dg_{t,\ell} - \sum_{t,\ell} I_{1_{t,\ell}}^2$$

$$= \sum_{t,\ell}\int_0^1 \log^2\left(\hat{C}_{t,\ell} + 2(1-\hat{C}_{t,\ell})g_{t,\ell}\right) dg_{t,\ell} - \sum_{t,\ell} I_{1_{t,\ell}}^2$$

$$= \sum_{t,\ell}\left(I_{3_{t,\ell}} - I_{1_{t,\ell}}^2\right)$$

### D.5 PROOF OF THEOREM 10

From Definition 3 we have: $\text{FPR}(\tau(\epsilon)) = 1 - CDF_{\neg w}(\tau(\epsilon)) = \epsilon$.

Additionally we know that the CDF of Bayesian score is as follows:

$$CDF_{\neg w}(\tau(\epsilon)) = \Phi\left(\frac{\ln\left(\frac{\tau(\epsilon)}{\alpha(1-\tau(\epsilon))}\right) - \mathbb{E}[BS(x)|\neg w]}{\sqrt{\text{Var}[BS(x)|\neg w]}}\right),$$

Hence we have:

$$1 - \epsilon = \Phi\left(\frac{\ln\left(\frac{\tau(\epsilon)}{\alpha(1-\tau(\epsilon))}\right) - \mathbb{E}[BS(x)|\neg w]}{\sqrt{\text{Var}[BS(x)|\neg w]}}\right),$$

Therefore:

$$\tau(\epsilon) = 1 - \frac{1}{1 + \alpha \exp\left(\mathbb{E}[\text{BS}(x)|\neg w] + \Phi^{-1}(1-\epsilon)\sqrt{\text{Var}(\mathbb{E}[BS(x)|\neg w])}\right)}$$

### D.6 PROOF OF THEOREM 11

By applying Proposition 2, and the detection threshold $\tau(\epsilon)$ in Theorem 10, we have

$$\mathbb{E}[\text{TPR}(\tau(\epsilon))|\text{FPR} = \epsilon] = 1 - CDF_{\text{BS}(x)|w}(\tau(\epsilon)) = 1 - \Phi\left(\frac{\ln\left(\frac{\tau(\epsilon)}{\alpha(1-\tau(\epsilon))}\right) - \mathbb{E}[BS(x)|w]}{\sqrt{\text{Var}[BS(x)|w]}}\right)$$

$$= 1 - \Phi\left(\frac{\mathbb{E}[\text{BS}(x)|\neg w] + \Phi^{-1}(1-\epsilon)\sqrt{\text{Var}[\text{BS}(x)|\neg w]} - \mathbb{E}[\text{BS}(x)|w]}{\sqrt{\text{Var}[\text{BS}(x)|w]}}\right)$$

### D.7 PROOF OF COROLLARY 3

From Proposition 2 and threshold given in Theorem 10 we have:

$$\mathbb{E}[\text{TPR}(\epsilon)|\text{FPR} = \epsilon] = 1 - \Phi\left(\frac{\mathbb{E}[\text{BS}(x)|\neg w] + \Phi^{-1}(1-\epsilon)\sqrt{\text{Var}[\text{BS}(x)|\neg w]} - \mathbb{E}[\text{BS}(x)|w]}{\sqrt{\text{Var}[\text{BS}(x)|w]}}\right)$$

We now analyze each term in this equation which is the sum of independent $\{g_{t,\ell}\}$'s. For simplicity, we only analyze a single $g_{t,\ell}$.

**Bernoulli(0.5)**: For $g_{t,\ell} \sim \text{Bernoulli}(0.5)$, and from Theorem 8 we have:

$$\mathbb{E}[g_{t,\ell}|\neg w] = \frac{1}{2}\log\left(\frac{1+\hat{C}_{t,\ell}}{2}\right) + \frac{1}{2}\log\left(\frac{3-\hat{C}_{t,\ell}}{2}\right)$$

$$\text{Var}[g_{t,\ell}|\neg w] = \frac{1}{2}\cdot\log^2\left(\frac{3-\hat{C}_{t,\ell}}{2}\right) + \frac{1}{2}\cdot\log^2\left(\frac{\hat{C}_{t,\ell}+1}{2}\right) - \left[\frac{1}{2}\cdot\log\left(\frac{3-\hat{C}_{t,\ell}}{2}\right) + \frac{1}{2}\cdot\log\left(\frac{1+\hat{C}_{t,\ell}}{2}\right)\right]^2$$

These values are constant within all g-values. Since these terms are strictly ascending or descending w.r.t. $\hat{C}_{t,\ell}$ we have:

$$\frac{1}{2}\log\frac{3}{4} \le \mathbb{E}[g_{t,\ell}|\neg w] \le 0; \quad 0 \le \text{Var}[g_{t,\ell}|\neg w] \le \frac{1}{4}(\log 3)^2$$

Therefore, as the number of layers increases, the (negative) expected value of unwatermarked Bayesian score decreases and the (positive) variance increases.

Additionally, from Theorem 8 we have:

$$\mathbb{E}[g_{t,\ell}|w] = \frac{1+C_{t,\ell}}{4}\log\left(\frac{1+\hat{C}_{t,\ell}}{2}\right) + \frac{3-C_{t,\ell}}{4}\log\left(\frac{3-\hat{C}_{t,\ell}}{2}\right)$$

$$\text{Var}[g_{t,\ell}|w] = \frac{3 - C_{t,\ell}}{4} \cdot \log^2\left(\frac{3 - \hat{C}_{t,\ell}}{2}\right) + \frac{1 + C_{t,\ell}}{4} \cdot \log^2\left(\frac{\hat{C}_{t,\ell} + 1}{2}\right)$$

$$- \left[\frac{3 - C_{t,\ell}}{4} \cdot \log\left(\frac{3 - \hat{C}_{t,\ell}}{2}\right) + \frac{1 + C_{t,\ell}}{4} \cdot \log\left(\frac{1 + \hat{C}_{t,\ell}}{2}\right)\right]^2$$

All these terms are strictly descending or ascending w.r.t. $C_{t,\ell}$. Hence, the maximum happens at end points of the intervals which is 0 and 1. Therefore, we have:

$$0 \le \mathbb{E}[g_{t,\ell}|w] \le \frac{1}{4}\log\left(\frac{1 + \hat{C}_{t,\ell}}{2}\right) + \frac{3}{4}\log\left(\frac{3 - \hat{C}_{t,\ell}}{2}\right)$$

$$0 \le \text{Var}[g_{t,\ell}|w] \le \frac{3}{4} \cdot \log^2\left(\frac{3 - \hat{C}_{t,\ell}}{2}\right) + \frac{1}{4} \cdot \log^2\left(\frac{\hat{C}_{t,\ell} + 1}{2}\right) - \left[\frac{3}{4} \cdot \log\left(\frac{3 - \hat{C}_{t,\ell}}{2}\right) + \frac{1}{4} \cdot \log\left(\frac{1 + \hat{C}_{t,\ell}}{2}\right)\right]^2$$

We can see that both terms are positive. Hence, as the number of layer increases, both the variance and expected value of watermarked Bayesian score increase.

Now that we have constant upper bound and lower bound for each term, for ease of description, we use the big-O notation to describe them:

$$\mathbb{E}[\text{BS}(x)|\neg w] = \sum_{t,\ell} \mathbb{E}[g_{t,\ell}|\neg w] = -\Theta(m), \quad \sqrt{\text{Var}[\text{BS}(x)|\neg w]} = \sqrt{\text{Var}\sum_{t,\ell}[g_{t,\ell}|\neg w]} = \Theta(\sqrt{m}),$$

$$\mathbb{E}[\text{BS}(x)|w] = \sum_{t,\ell} \mathbb{E}[g_{t,\ell}|w] = \Theta(m), \quad \sqrt{\text{Var}[\text{BS}(x)|w]} = \sqrt{\text{Var}\sum_{t,\ell}[g_{t,\ell}|w]} = \Theta(\sqrt{m}),$$

Hence, the behavior of the TPR function is:

$$\mathbb{E}[\text{TPR}(\epsilon)|\text{FPR} = \epsilon] = 1 - \Phi\left(\frac{-\Theta(m) + \Phi^{-1}(1 - \epsilon)\Theta(\sqrt{m}) - \Theta(m)}{\Theta(\sqrt{m})}\right)$$

$$= 1 - \Phi(-\Theta(\sqrt{m})) = 1 - e^{-\Theta(\sqrt{m})}$$

This indicates that as the number of layers m increases, the TPR increases.

**Uniform[0,1]**: For $g_{t,\ell} \sim \text{Uniform}(0, 1)$, and from Theorem 9, we have:

$$\mathbb{E}[g_{t,\ell}|\neg w] = I_{1_{t,\ell}}, \quad \text{Var}[g_{t,\ell}|\neg w] = I_{3_{t,\ell}} - I_{1_{t,\ell}}^2$$

These values are constant within all g-values. Since these terms are strictly ascending and descending according to $\hat{C}_{t,\ell}$ we have:

$$\log(2) - 1 \le \mathbb{E}[g_{t,\ell}|\neg w] \le 0, \quad \text{Var}[g_{t,\ell}|\neg w] = I_{3_{t,\ell}} - I_{1_{t,\ell}}^2 \approx 1$$

Therefore, as the number of layers increases the expected value of unwatermarked Bayesian score decreases and the variance increases.

Additionally, from Theorem 9 we have:

$$\mathbb{E}[g_{t,\ell}|w] = C_{t,\ell} \cdot I_{1_{t,\ell}} + 2(1 - C_{t,\ell})I_{2_{t,\ell}}$$

$$\text{Var}[g_{t,\ell}|w] = C_{t,\ell} \cdot I_{3_{t,\ell}} + 2(1 - C_{t,\ell})I_{4_{t,\ell}} - (C_{t,\ell} \cdot I_{1_{t,\ell}} + 2(1 - C_{t,\ell})I_{2_{t,\ell}})^2$$

All these terms are strictly descending or ascending according to $C_{t,\ell}$. Hence, the maximum happens at end points of the intervals which is 0 and 1. Therefore, we have:

$$I_{1_{t,\ell}} \le \mathbb{E}[g_{t,\ell}|w] \le 2I_{2_{t,\ell}}$$

$$0 \le 2(I_{4_{t,\ell}} - I_{2_{t,\ell}}^2) \le \text{Var}[g_{t,\ell}|w] \le I_{3_{t,\ell}} - I_{1_{t,\ell}}^2 \approx 1$$

Therefore we have:

$$-0.5 \le I_{1_{t,\ell}} - 2I_{2_{t,\ell}} \le \mathbb{E}[\text{BS}(x)|\neg w] - \mathbb{E}[\text{BS}(x)|w] \le 0$$

Hence, as the number of layer increases, the variance and expected value of watermarked Bayesian score increases.

Similar to the analysis on Bernoulli distribution, as each term has constant upper bound and lower bound, we can use the big-O notation:

$$\mathbb{E}[\text{BS}(x)|\neg w] - \mathbb{E}[\text{BS}(x)|w] = \Theta(-m),$$

$$\sqrt{\text{Var}[\text{BS}(x)|\neg w]} = \Theta(\sqrt{m}), \quad \sqrt{\text{Var}[\text{BS}(x)|w]} = \Theta(\sqrt{m}),$$

Hence, the behavior of the TPR function is:

$$\mathbb{E}[\text{TPR}(\epsilon)|\text{FPR} = \epsilon] = 1 - \Phi\left(\frac{\Phi^{-1}(1-\epsilon)\Theta(\sqrt{m}) - \Theta(m)}{\Theta(\sqrt{m})}\right)$$

$$= 1 - \Phi(-\Theta(\sqrt{m})) = 1 - e^{-\Theta(\sqrt{m})}$$

indicating that as the number of layers $m$ increases, the TPR increases.

## D.8 PROOF OF COROLLARY 4

We show that when $\hat{C}_{t,\ell}$ becomes 1, TPR does not change and it will saturate.

**Bernoulli(0.5):** If $\hat{C}_{t,\ell} = 1$, then:

$$\mathbb{E}[g_{t,\ell}|\neg w] = \frac{1}{2}\log\left(\frac{1+\hat{C}_{t,\ell}}{2}\right) + \frac{1}{2}\log\left(\frac{3-\hat{C}_{t,\ell}}{2}\right) = \frac{1}{2}\log(1) + \frac{1}{2}\log(1) = 0$$

$$\text{Var}[g_{t,\ell}|\neg w] = \frac{1}{2}\cdot\log^2\left(\frac{3-\hat{C}_{t,\ell}}{2}\right) + \frac{1}{2}\cdot\log^2\left(\frac{\hat{C}_{t,\ell}+1}{2}\right) - \left[\frac{1}{2}\cdot\log\left(\frac{3-\hat{C}_{t,\ell}}{2}\right) + \frac{1}{2}\cdot\log\left(\frac{1+\hat{C}_{t,\ell}}{2}\right)\right]^2$$

$$= \frac{1}{2}\cdot\log^2(1) + \frac{1}{2}\cdot\log^2(1) - \left[\frac{1}{2}\cdot\log(1) + \frac{1}{2}\cdot\log(1)\right]^2 = 0$$

$$\mathbb{E}[g_{t,\ell}|w] = \frac{1+C_{t,\ell}}{4}\log\left(\frac{1+\hat{C}_{t,\ell}}{2}\right) + \frac{3-C_{t,\ell}}{4}\log\left(\frac{3-\hat{C}_{t,\ell}}{2}\right) = \frac{1+C_{t,\ell}}{4}\log(1) + \frac{3-C_{t,\ell}}{4}\log(1) = 0$$

$$\text{Var}[g_{t,\ell}|w] = \frac{3-C_{t,\ell}}{4}\cdot\log^2(1) + \frac{1+C_{t,\ell}}{4}\cdot\log^2(1) - \left[\frac{3-C_{t,\ell}}{4}\cdot\log(1) + \frac{1+C_{t,\ell}}{4}\cdot\log(1)\right]^2 = 0$$

Therefor since for all $\ell > M$ we have $\hat{C}_{t,\ell} = 1$ ($\hat{C}_{t,\ell}$ is non-decreasing w.r.t. $l$), the TPR remains constant after $\ell > M$.

**Uniform[0,1]:** Similar to Bernoulli(0.5), we can simply prove that all $I_{1_{t,\ell}}, I_{2_{t,\ell}}, I_{3_{t,\ell}}, I_{4_{t,\ell}}$ become constant after $\ell > M$ for an existing M when $\hat{C}_{t,\ell}$ becomes one for the first time for all t.

## D.9 PROOF OF THEOREM 12

Let the $g$-value distribution $f_g$ be Bernoulli($p$) for some $0 < p < 1$. According to Theorem 17, the watermarked $g$-value distribution is Bernoulli$\big(p + p(1-p)(1-C_{t,\ell})\big)$.

We first calculate the threshold, expected value and variance of both unwatermarked and watermarked Mean Score. Similar to proofs for Theorems 3 and 5, we have

$$\mathbb{E}[\text{MS}(x)|\neg w] = p \quad \text{Var}[\text{MS}(x)|\neg w] = \frac{1}{mT}p(1-p)$$

$$\mathbb{E}[\text{MS}(x)|w] = \frac{1}{mT}\sum_{t,\ell} p + p(1-p)(1-C_{t,\ell})$$

$$\text{Var}[\text{MS}(x)|w] = \left(\frac{1}{mT}\right)^2 \sum_{t,\ell} (p + p(1-p)(1-C_{t,\ell}))\,(1 - (p + p(1-p)(1-C_{t,\ell})))$$

Then we can write the TPR as follows:

$$\mathbb{E}[\text{TPR}(\epsilon)|\text{FPR} = \epsilon] = 1 - \Phi\left(\frac{\mathbb{E}[\text{MS}(x)|\neg w] + \Phi^{-1}(1-\epsilon)\sqrt{\text{Var}[\text{MS}(x)|\neg w]} - \mathbb{E}[\text{MS}(x)|w]}{\sqrt{\text{Var}[\text{MS}(x)|w]}}\right)$$

$$= 1 - \Phi\left(\frac{p + \frac{\Phi^{-1}(1-\epsilon)\sqrt{p(1-p)}}{\sqrt{mT}} - \left(p + \frac{1}{mT}\sum_{t,\ell} p(1-p)(1-C_{t,\ell})\right)}{\frac{1}{mT}\sqrt{\sum_{t,\ell}\left(p + p(1-p)(1-C_{t,\ell})\right)\left(1 - \left(p + p(1-p)(1-C_{t,\ell})\right)\right)}}\right)$$

$$= 1 - \Phi\left(\frac{\sqrt{mT}\Phi^{-1}(1-\epsilon)\sqrt{p(1-p)} - \sum_{t,\ell} p(1-p)(1-C_{t,\ell})}{\sqrt{\sum_{t,\ell}\left(p + p(1-p)(1-C_{t,\ell})\right)\left(1 - \left(p + p(1-p)(1-C_{t,\ell})\right)\right)}}\right)$$

$$= 1 - \Phi\left(Z(p)\right),$$

We want to minimize $Z(p)$.

**Step 1: Simplify the Argument** $Z(p)$**.** Let $S = \sum_{t,\ell}(1 - C_{t,\ell})$ and $V(p) = \sum_{t,\ell}\left(p + p(1-p)(1-C_{t,\ell})\right)(1 - p - p(1-p)(1-C_{t,\ell}))$. Then:

$$Z(p) = \frac{\sqrt{mT}\Phi^{-1}(1-\epsilon)\sqrt{p(1-p)} - p(1-p)S}{\sqrt{V(p)}}.$$

**Step 2: Asymptotic Approximation for Large** $mT$**.** For large $mT$, approximate $V(p)$ using the law of large numbers: $V(p) \approx mT \cdot \mathbb{E}\left[(p + p(1-p)(1-C))(1-p-p(1-p)(1-C))\right]$, where the expectation is over the distribution of $C_{t,\ell}$.

Let $\overline{C} = \mathbb{E}[C_{t,\ell}]$. Then: $V(p) \approx mT\left(p(1-p) + O(p^2(1-p)^2)\right)$. Thus for large $mT$, the denominator behaves like:

$$\sqrt{V(p)} \approx \sqrt{mTp(1-p)},$$

**Step 3: Approximate** $Z(p)$ **for Large** $mT$**.** Substitute $\sqrt{V(p)}$:

$$Z(p) \approx \frac{\sqrt{mT}\Phi^{-1}(1-\epsilon)\sqrt{p(1-p)} - p(1-p)S}{\sqrt{mTp(1-p)}} = \Phi^{-1}(1-\epsilon) - \frac{S\sqrt{p(1-p)}}{\sqrt{mT}}$$

**Step 4: Minimizing** $Z(p)$ **to Maximize TPR.** As $\Phi$ is monotonically increasing, minimizing $Z(p)$ maximizes $1 - \Phi(Z(p))$. The dominant term is:

$$Z(p) \approx \Phi^{-1}(1-\epsilon) - \frac{S}{\sqrt{mT}}\sqrt{p(1-p)}.$$

To minimize $Z(p)$, maximize $\sqrt{p(1-p)}$, which peaks at $p = \frac{1}{2}$:

$$p^* = \arg\max_{p\in(0,1)} \sqrt{p(1-p)} = \frac{1}{2}.$$

