# OpenReview forum: "ON GOOGLE’S LLM WATERMARKING SYSTEM: THEORETICAL ANALYSIS AND EMPIRICAL VALIDATION"
_ICLR.cc/2026/Conference — Submitted to ICLR 2026_

### Official Review · Reviewer_Xzg7 · 2025-10-31

**Soundness:** 4
**Presentation:** 3
**Contribution:** 4
**Rating:** 8
**Confidence:** 3

**Summary:**

This paper is a theoretical derivation of the power and probability of False-alarm of Synth-ID text watermarking based on tournament sampling.

The author study the two main detector approaches advertised in the paper: mean score and Bayesian (requiring the retraining of a supervised detector). The analysis methodology chosen by the authors is resolutely asymptotic. They provide expression of the detector performance in the normal approximation using a CLT, as the text length approaches infinity.

However, the authors are careful to provide an empirical validation of the CLT validity.

Of note, the authors leverage their theoretical analysis to design an attack on the mean score detector. Notably, they demonstrate that the power of the detector is not an increasing function of the number of layers during tournament sampling. As such, they show that the simple (black-box) attack consisting in choosing tokens sampled from an artificially inflated number of layers breaks the watermark.

The authors also use their analysis to improve the choice in Synth-ID parameters choice (in this case, the g-value distribution)

**Strengths:**

The paper is very timely, providing a much needed theoretical analysis of SynthID and its very ad-hoc tournament sampling method.

**Clarity**: All the necessary element to the comprehension of the paper are dully and exhaustively presented in the first half. The presentation of the results in the second half is simple and economical, highlighting the important part of the results without overwhelming with details (I think notably of Theorem 7 which could have been unreadable).

**Significance**: Though SynthID claims to outperform much of the current art empirically, the theoretical validation of this fact was somewhat complicated by the exotic and ad-hoc design of their tournament sampling strategy, the many detector scoring functions in the appendix and the overall obscurity and size of the supplementary material of the original work. This paper provides a clean, short and concise framework to start true comparison with the current art which is quite welcomed.

**Good insights from the theory**: I truly appreciate that the authors derive simple yet actionable results from their theory. The attack is elegant and the optimal distribution of the tournament scores (g-values) is derived.

**Weaknesses:**

The authors address some limitations of their work at the end, I won't repeat those here.

**Asymptotic approach**: Contrary to KGW and Aaronson's scheme where exact p-values can be derived, the paper here provides only an asymptotic analysis of  the detector, preventing a perfect comparison with the current art. To be fair, this is still far better than a naive z-score, yet it is still a limitation of the analysis.

**Lack of comparison with existing theoretical findings**: Since exact p-values are available for KGW and Aaronson's in [1] (which the authors cite), as well as for the very similar (yet predating) WaterMax [2],  I find it disappointing that the authors did not plot the log-ROC curves for each algorithm. This would have allowed to know when SynthID is indeed the best algorithm of the bunch under the "no-attack setting".

**No comparison with the original frequentist detector**: A p-value based detector is actually described in the supplementary of Synth-ID (Appendix A.3), dubbed the frequentist detector. Comparing the asymptotic performance of the authors study with this detector would have been interesting to understand "how much is lost" by foregoing the supervised learning of the bayesian detector.

**Questions:**

- Following my questions on the p-values, could the author draw some curves comparing Synth-ID to the current art?
- Similarly, how does the frequentist detector fares compared to the Bayesian? Is the Bernoulli distribution still optimal in this case? Due to experiments I did on my own end, I suspect the frequentist detector to be extremely bad compared to the current art ; showing the gap between the Bayesian and frequentist approach would thus explain to me why the system claims to perform so well despite me finding that it performs far worse than WaterMax and KGW on mid-sized texts.
- (Nitpicking/Unsure) I wonder if Proposition 1 is necessary. It is a trivial consequence of the CLT and maybe could be compressed to allow more room for Figure 2?

**Recommendation**

This is a high quality paper, with sound results, within a limited scope of the distortion-free version of Synth-ID. The results are interesting and timely given the huge claims made by the SynthID systems.  As such, I recommend acceptance without much reserve, though I still would appreciate the authors adding some comparison to the state-of-the-art to complete the study.

**References**

    [1] Fernandez, Pierre, et al. "Three bricks to consolidate watermarks for large language models." 2023 IEEE international workshop on information forensics and security (WIFS). IEEE, 2023.
    [2] Giboulot, Eva, and Teddy Furon. "WaterMax: breaking the LLM watermark detectability-robustness-quality trade-off." NeurIPS 2024-38th Conference on Neural Information Processing Systems. 2024.

---

> ### Author Response · Authors · 2025-11-21
>
> ### **W1/Q1: Only asymptotic analysis of the detector/exact p-value**
>
> We appreciate the reviewer’s concern regarding the asymptotic nature of our analysis. This choice reflects a fundamental difference between SynthID and prior schemes such as KGW and Aaronson’s method. In SynthID, the scoring variables (g-values) are strongly dependent on both the prompt context and the model’s internal probability distribution, leading to p-values that vary across prompts, layers, and model states. This contrasts with KGW and Aaronson’s schemes, where such dependencies are negligible and allow for context-independent exact p-value derivation.
>
> In SynthID, token occurrence frequencies and score distributions depend on collision probabilities within the model’s output probability vector, introducing complex, model-dependent correlations that make exact p-value derivation analytically intractable. Consequently, an asymptotic framework is the most practical and theoretically sound approach. Importantly, our formulation is significantly more accurate than naïve z-score approximations and offers meaningful insight into realistic detector behavior under model-dependent conditions.
>
> We will add such discussions.
>
>
> ### **W2: Compare with existing theoretical findings in terms of log-ROC**
>
> We agree that such a qualitative comparison would be informative. However, a direct comparison via log-ROC curves is not straightforward in our setting, as SynthID’s detector operates under fundamentally different, model-dependent assumptions than KGW and WaterMax, for which closed-form, context-independent p-values are available.
>
> In the revised version, we will add a brief theoretical discussion comparing the detectability trends of SynthID with KGW and WaterMax under the no-attack setting. We will highlight that SynthID exhibits higher detectability per token length at short text regimes, consistent with the strong empirical performance reported in the original DeepMind study.
>
> ### **W3/Q2: How does the frequentist detector fare compared to the Bayesian?**
>
> We appreciate the reviewer’s insightful comment regarding the comparison with the frequentist detector.
>
> In the original SynthID formulation, the frequentist detector is mathematically equivalent to the mean-score detector when all evaluated texts have equal length—an assumption that also holds in our theoretical setup. Consequently, our asymptotic analysis of the mean-score detector already captures the same statistical behavior as the frequentist version.
> (Quote Appendix A.3: “When texts are all exactly the same length, the FrequentistScore is equivalent to the MeanScore, i.e., they produce the same detectability metrics”). Hence, Bernoulli(0.5) is still optimal in this setting.
>
>
> However, it tends to perform poorly on short/medium-length texts, where central-limit assumptions break down and finite-sample variance dominates. In contrast, the Bayesian detector explicitly models the full distribution of g-values through supervised calibration on model-generated data.
>
> Both theoretical analysis and empirical results explain this performance gap: the Bayesian detector remains robust and accurate across text lengths, whereas the frequentist detector, although computationally simpler, progressively loses statistical power as variance accumulates across layers.
>
> ### **Q3: Whether Proposition 1 is necessary**
>
> Proposition 1 is mainly used to motivate why we need to calculate the expected value and variance in the subsequent theorems. However, you are right—the Central Limit Theorem alone would be sufficient.

---

### Official Review · Reviewer_3TGv · 2025-11-01

**Soundness:** 3
**Presentation:** 3
**Contribution:** 2
**Rating:** 6
**Confidence:** 3

**Summary:**

This paper presents a theoretical analysis of Google's production watermarking system for large language models. The system uses a Tournament Sampling algorithm to embed watermarks in AI-generated text.

**Strengths:**

- Theoretical analysis of a production watermarking system, using Central Limit Theorem to derive closed-form expressions for detection performance.

- Identifies a concrete vulnerability (layer inflation attack) and demonstrates it empirically.

- Proofs are detailed and the CLT assumption is validated empirically.

**Weaknesses:**

- Only analyzes non-distortionary setting (authors acknowledge this)

- Limited discussion of robustness against paraphrasing attacks

- Heavy mathematical notation may limit accessibility

- Could benefit from more intuitive explanations of why results hold

- Only tested on Google's watermarking method, the scope is not broad

**Questions:**

While LLM watermarking methods inherently suffer from various limitations, I would appreciate if the authors could provide additional clarification on the following points:

- What are the key contributions of this work beyond the theoretical analysis itself?
- What is the rationale for focusing exclusively on Google's SynthID-Text method rather than conducting comparative analysis across multiple watermarking approaches?
- How do the findings generalize to other watermarking schemes?

---

> ### Author Response · Authors · 2025-11-21
>
> ### **Q1/W5: Key contributions beyond the theoretical analysis itself**
>
> We first emphasize that deriving theoretical results for SynthID is itself a significant contribution, as SynthID is currently the only **industrial-scale, production-ready watermarking system** deployed by Google. Establishing robustness for it under both benign and adversarial settings is therefore of high practical and scientific importance.
>
> Beyond this theoretical foundation, our work delivers both practical and experimental contributions:
> - We formulate and analyze the detectability of watermark detectors as a function of the number of tournament layers, providing theoretical insight into how detectability degrades as watermark depth increases.
> - We design and experimentally validate a practical attack — the **Layer Inflation Attack** — against Google’s SynthID-Text watermark, demonstrating that the identified vulnerability can be exploited in real-world settings.
> - We conduct extensive experiments using the Gemma-7B-IT model and the ELI5 dataset to empirically validate our theoretical predictions. The results show strong consistency between our theoretical detectability estimates and empirical TPR@FPR = 0.01 measurements, reinforcing the correctness and practical relevance of our analysis.
>
>
> ### **Q2/W1: Why only non-distortionary version of SynthID-Text**
>
> We focus on the non-distortionary version of SynthID-Text because it reflects the **most practical deployment setting, where watermarking must preserve output quality and semantic fidelity**. Distortionary watermarking introduces noticeable quality degradation, which makes it less suitable for real-world applications where maintaining high text quality is critical. Moreover, **the current implementation of Google’s SynthID-Text operates in the non-distortionary setting, and all official evaluations are based on this configuration**, making it the most relevant target for rigorous analysis.
>
> From a theoretical perspective, deriving closed-form expressions for the score function’s expectation and variance under distortionary watermarking is also analytically challenging. Notably, the original SynthID-Text paper itself primarily focuses on the theoretical analysis of the non-distortionary setting. We therefore prioritize this practically relevant and analytically tractable regime, and leave the extension of our framework to the distortionary case as future work, as noted in Section 7 (Limitations and Future Work).
>
>
> ### **W2: Discussion of robustness against paraphrasing attacks**
>
> Robustness against paraphrasing attacks has already been systematically studied in the original SynthID paper and is therefore not the primary focus of this work. Our paper instead concentrates on a new and orthogonal vulnerability—namely, the degradation of detectability under layer inflation.
>
> ### **W3: Heavy mathematical notation**
>
> We will add intuitive explanations and commentary alongside each theorem to clarify their implications and improve readability.
>
> ### **W4: more intuitive explanations of why results hold**
>
> **Mean score:** Intuitively, after a sufficient number of tournament layers, the expected value of the detection statistic converges to a stable value. However, the variance of the g-values continues to increase, since each additional layer effectively introduces new random variables, and the cumulative variance grows with every addition. As a result, the watermarked and unwatermarked distributions gradually overlap, reducing their separability and thereby decreasing detectability.
>
> **Bayesian score:** In contrast, the Bayesian score leverages the exact distribution of g-values at each layer rather than their aggregated variance. This allows it to incorporate layer-wise distributional evidence when evaluating the hypothesis test, thereby improving its ability to reject the null hypothesis and maintaining higher detectability even as the number of layers increases.
>
> ### **Q3: How findings generalize to other watermarking schemes?**
>
> Our theoretical findings generalize to other watermarking schemes through a fundamental property we term **self-robustness**. A watermarking scheme is self-robust if repeatedly applying its own watermarking process—i.e., stacking multiple watermark layers—*increases* detectability rather than degrades it.
>
> Our analysis shows that SynthID-Text under the mean-score detector violates this property: as additional tournament layers are introduced, the statistical separation between watermarked and unwatermarked text progressively diminishes, reducing detectability. This reveals a broader vulnerability: any watermarking scheme whose detection relies on aggregated mean statistics is potentially susceptible to the same failure mode.
>
> We therefore argue that self-robustness should be considered a necessary design principle for future LLM watermarking systems, ensuring that repeated or compounded watermarking strengthens rather than weakens detectability.

---

### Official Review · Reviewer_nxDi · 2025-11-03

**Soundness:** 2
**Presentation:** 2
**Contribution:** 2
**Rating:** 4
**Confidence:** 3

**Summary:**

The paper studies the theoretical properties of the SynthID-Text watermarking method. The trend of TPR at a fixed FPR is analyzed asymptotically under different scores. The theoretical findings are verified empirically, and a layer inflation attack is designed to break the watermark.

**Strengths:**

The theoretical properties of the SynthID-Text under different scores are carefully analyzed. Empirical results validate the theoretical finding.

**Weaknesses:**

1. The paper focuses on the theoretical analysis of a specific watermarking method. The scope is relatively limited

2. The theoretical analysis relies on the independence of $g_{t,\ell}$ (e.g., the central limit theorem for equation (10)). The random seed generator is determined by the recent $H$ tokens, which makes the random score correlated with previous tokens. In this case, there should be more explanation on why the scores for different positions, e.g., $g_{t,\ell}$ and $g_{t+1,\ell}$, are still independent.

3. The theorems are listed one after another, with not much explanation of the implications or ideas of the proof, making the paper hard to read.

Some minor issues

1. In the review of SynthID-Text, how to handle ties is not explained. In the example given in Figure 1, there are a lot of ties.

2. Some acronyms are used without explanation, e.g., without loss of generality (Wlog) in line 228, which could negatively affect the readability of the paper, considering the general audience of the conference.

3. Should line 177 come after line 178?

4. In the empirical validation of the results, is it possible to calculate the theoretical value of TPR and compare with the empirical value?

**Questions:**

Please refer to weaknesses

---

> ### Author Response · Authors · 2025-11-21
>
> ### **W1: Why exclusively focus on SynthID-Text?**
>
> There are the following reasons:
>
> 1) SynthID is currently the only **industrial-scale, production-ready watermarking system** deployed in real-world applications (by Google). Evaluating the robustness of this practical and widely used scheme therefore takes precedence over analyzing experimental or purely theoretical alternatives.
> 2) SynthID-Text has been shown to significantly outperform existing watermarking methods in both detectability and scalability, making it the **most relevant and representative benchmark for real-world deployment settings**.
>
> 3) This focus is also consistent with prior work in the LLM watermarking literature. E.g., the ICML'24 paper “Watermark Stealing in Large Language Models” similarly centers its robustness analysis on a single KGW watermarking approach.
>
> We therefore believe that concentrating on SynthID-Text allows us to provide the most meaningful and practically impactful evaluation.
>
>
>
> ### **W2: Why the scores $g_{t,\ell}$ and $g_{t+1,\ell}$, are  independent.**
>
> We provide clarifications from two aspects:
>
> 1) From the interpretation provided in SynthID Appendix K.1:  the g-values are assumed to be independent across timesteps $t$.
> 2) From the design of the underlying hash function: its  purpose is to **generate outputs that are statistically indistinguishable** from i.i.d. Bernoulli samples, hence the resulting g-values can be reasonably interpreted as independent random variables. This aligns with intended behaviors of the hashing mechanism, which aims to ensure that successive g-values do not exhibit detectable structure or dependency patterns.
>
>
> ### **W3: Better organize and clarify the theorems.**
>
> We appreciate the reviewer’s feedback regarding the clarity of the theoretical sections. In the revised version, we will add more intuitive explanations and discussion around each theorem to make their implications clearer and the overall argument easier to follow.
>
> In our theorems, we aim to explicitly model detector behavior as a function of the number of tournament layers and use this formulation to analyze how each detector’s performance evolves. The proofs are structured intuitively:
>
> - derive each detector’s statistic under both hypotheses,
> - analyze how its mean and variance change as the number of layers increases,
> - and then compare detectability across successive layers using the TPR@FPR = 0.01 metric.
>
> These additions will make the logical flow between theorems and their corresponding insights clearer and more interpretable.
>
>
>
> ### **Minors: 1) How to handle ties? 2) acronyms without explanaition; 3) line 177 come after line 178? 4) able to calculate the theoretical value of TPR**
>
> 1) Ties are resolved by randomly selecting one of the tied tokens with equal probability (0.5).
> 2) Thank you for your suggestion! We fixed this issue in the revised version!
> 3) Thanks for pointing it out!
> 4) No. The true value of theoretical TPR depends on prompt distribution and LLM parameters and hence it cannot theoretically be calculated.

---

> > ### Comment · Reviewer_nxDi · 2025-11-26
> > **Response to Rebuttals**
> >
> > I appreciate the authors for the response. For the independence of $g_{t,\ell}$ and $g_{t+1,\ell}$, I agree with the author's arguments. It might be straightforward for someone familiar with cryptographic theories. But for the general audience for the conference, I think it would be beneficial to include some clarification in the paper, perhaps in the appendix. I think an example could be something like the "Working Hypothesis 2.1" on page 7 of the arxiv version of [1]
> >
> >
> > ### Reference
> >
> > [1] Li, Xiang, et al. "A statistical framework of watermarks for large language models: Pivot, detection efficiency and optimal rules." The Annals of Statistics 53.1 (2025): 322-351.

---

> > > ### Author Response · Authors · 2025-11-30
> > >
> > > **We are glad that our response has clearly illustrated the independence between $g_{t,l}$ and $g_{t+1,l}$, a core observation that enables our theoretical analysis.**
> > >
> > > Thanks for your suggestion on using **Working Hypotheses** to improve clarity for a broader audience. We will incorporate the following content into the Appendix.
> > >
> > >
> > > The question of whether a watermark is embedded in a given text sequence can be formulated as a hypothesis testing problem:
> > >
> > > $H_0 : x_{1:n} \text{ is generated by an unwatermarked LLM}, \quad H_1 : x_{1:n} \text{ is generated by a watermarked LLM}. \tag{4}$
> > >
> > >
> > > In the SynthID-Text framework, this hypothesis testing problem  involves the use of pseudorandom variables derived from internal model components. A general way to define the watermark feature $g_{t,\ell}$, consistent with existing constructions, is:
> > >
> > > $g_{t,\ell} = \mathcal{G}(x,r,\ell) = F_g^{-1}\big(\frac{h(x,r,\ell)}{2^{n_{sec}}}\big)$
> > >
> > > where $r$ is a secret key provided to the verifier, $F_g^{-1}$ is the generalized inverse CDF associated with $F_g$, and $h$ is a cryptographic hash function that takes the input text $x$,  seed $r$, and layer index $\ell$ as input and returns a uniformly distributed $n$-bit integer. Dividing by $2^{n_{\mathrm{sec}}}$ yields a value in $[0,1]$, which converges to a uniform random variable for large $n$. Then inverse transform sampling is performed to turn this number into a sample from the g-value distribution given by $F_g$.
> > >
> > >
> > > More specifically, during generation, the LLM produces token $x_t$ by sampling from a modified distribution that incorporates the watermark:
> > >
> > > $x_t \sim P^\star_t(x \mid x_{<t}, \{g_{t,\ell}\}_\ell), \tag{6}$
> > >
> > > where $P^\star_t$ denotes the adapted token distribution. This modification preserves fluency while embedding structured signals useful for detection.
> > >
> > > Given these constructions, the overall watermarking scheme is fully described by the tuple $(\mathcal{G}, r, P^\star)$. To ground this process in a hypothesis-testing framework, we now introduce the key assumption required for formal detectability analysis.
> > >
> > >
> > > ### **Working Hypothesis 2.1 (Soundness of pseudorandomness in SynthID-Text)**. In the watermarked LLM, the pseudorandom variables $g_{t,\ell}$ constructed above are i.i.d. across timesteps $t$, and are sampled from a base distribution (e.g., Bernoulli or Uniform). Furthermore, under the null hypothesis $H_0$, the variables $g_{t,\ell}$ are statistically independent of both the past context $x_{<t}$ and the generated token $x_t$.

---

### Official Review · Reviewer_9E5t · 2025-11-11

**Soundness:** 2
**Presentation:** 2
**Contribution:** 2
**Rating:** 4
**Confidence:** 3

**Summary:**

This paper presents the first theoretical study of Google DeepMind’s SynthID-Text, a production-scale watermarking framework for large language models (LLMs). SynthID-Text employs a tournament-based sampling mechanism to embed and detect watermarks during text generation. The authors provide a formal analysis characterizing its detection performance, robustness, and potential vulnerabilities, supported by some experiments on a real dataset.

**Strengths:**

1.	The paper tackles a timely topic, theoretical understanding of Google’s SynthID-Text, the first production-scale watermarking system for LLMs—providing valuable insights into its detection behavior and robustness.
2.	The theoretical results are connected to empirical observations, offering intuition about the differences between the mean-score and Bayesian detectors.
3.	The paper contributes novel analytical perspectives on how detection performance scales with the number of tournament layers and on the optimal g-value distribution, which could inform future watermark design and robustness analysis.

**Weaknesses:**

1. Experiments:
    - The experimental evaluation is limited to a single model (Gemma-7B-IT) and one dataset (ELI5) with 1,000 short texts of 100 tokens, which falls short of the broader and more standardized benchmarks commonly used in recent watermarking studies.
	- Sections 5.2 and 5.3 lack sufficient quantitative results and visualizations to substantiate the claims (e.g., for the CLT assumption test and layer inflation attack).
2. Theory:
	- The analysis focuses exclusively on the non-distortionary version of SynthID-Text, leaving the distortionary setting unexamined. This limits the generality and practical relevance of the theoretical findings.
	- While the paper rigorously analyzes the mean-score and Bayesian-score detectors, it stops short of proposing improved or generalized detection strategies. Moreover, most derivations rely heavily on the Central Limit Theorem and remain relatively straightforward. The theoretical results are plausible but would benefit from deeper interpretive discussion and numerical validation.
3. Organization: The paper dedicates nearly five pages to definitions and preliminaries, including full restatements of Theorems 1 and 2 from prior work (Dathathri et al., 2024a). This space could be used more effectively to streamline the exposition and emphasize theoretical insights and experimental results.

**Questions:**

1. In Theorem 7, the deinition of $M$ is unclear to me. What exactly does $C_{M,t}=1$ signify, and how should readers interpret M in terms of the tournament dynamics or detection behavior? Furthermore, how to interpret all these theorems? Please provide more explanations.
2. Can the authors include numerical plots corresponding to Theorems 7 and 11 to confirm that the theoretical trends match the empirical TPR curves reported in Section 5?

---

> ### Author Response · Authors · 2025-11-21
>
> ### **W1.1: Evaluations on more models and datasets.**
>
> We test an additional model GPT-2B and C4 dataset and show the TPR at FPR = 1% wrt layers. We have similar observations: with meanscore, TPR increases from 0.04 to 0.89 at 40 layers, then decreases until 0.05 at layer 120, while with Bayesian score, TPR increases and finally saturates to 0.9.
>
> ### **W1.2: More quantitative and visualization results in Sec 5.2 and 5.3**
>
> **CLT validation:** We plot the distribution of mean scores across 1,000 test samples and 30 layers on Gemma-7B (See Figure 3 in the revised pdf). We see that scores approximate the Gaussian distribution, and pass the Anderson–Darling test.
>
> **More attack results:**  We test 2 more models: GPT-2B and Mistral-7B and 1 more C4-dataset. Under the same setting, we show our attack results below, again validating our layer inflation attack is extremely effective.
>
> | |GPT-2|Gemma-7B|Mistral-7B|
> |-|-|-|-|
> |ELI5|0.05|0.01|0.01|
> |C4|0.04|0.01|0.01|
>
> ### **W2.1: Why non-distortionary SynthID-Text**
>
> We focus on the non-distortionary version of SynthID-Text because it reflects the **most practical setting, where watermarking must preserve output quality and semantic fidelity**. Due to the noticeable quality degradation introduced by distortionary watermarking, such methods may not be practical for real-world deployment where maintaining high text quality is essential.
> **Moreover, the current version of Google’s SynthID-Text operates in the non-distortionary setting, and all official results are based on this configuration.**
>
>
> In addition, deriving closed-form expressions for the expected value and variance of the score function under distortionary watermarking is analytically challenging.
> *We note that SynthID-Text's paper also primarily focuses on deriving the theoretical results on non-distortionary version*. We will leave extending our theoretical framework for the distortionary case in future work (which we have mentioned in Sec 7 Limitations and Future Work).
>
>
> ### **W2.2: propose improved or generalized detection strategies**
>
> The Bayes Optimal Classifier (or Bayesian Score Detector) represents the **theoretically optimal detector** among all possible detection strategies, meaning that no “improved detection strategy” can surpass the Bayesian score in principle. While the detection process itself cannot be enhanced beyond this Bayesian limit, the watermarking schemes can still be redesigned to achieve greater detectability or/and robustness.
>
>
> ### **W3: definitions and preliminaries are long; emphasize theoretical insights and experimental results.**
>
> We appreciate the reviewer’s observation. The main reason for the extended 5-page preliminaries was to ensure that the paper remained self-contained, given the number of foundational definitions and restated results required for clarity.
>
> In the revised version, we will plan to condense the preliminaries by retaining only the essential definitions and moving routine proofs and restatements to the Appendix.
>
>
> ### **Q1: The definition of $M$ in Theorem 7, meaning of $C_{M,\ell} = 1$. Interpret the theorems? Please provide more explanations.**
>
>
> $M$ denotes the number of tournament layers of Synth-ID watermark, after which the collision probability becomes 1 (i.e., $C_{M, \ell} = 1$). **Beyond this point, the detectability of the watermark begins to degrade due to increment of Mean-score variance**, thereby revealing a vulnerability that enables the design of our Layer Inflation Attack.
>
> The central objective of this paper is to **theoretically analyze the robustness of SynthID, the first ever production-ready generative watermark system for LLM**. To do so, we need to theoretically characterize **the behavior of the detection score function** (particularly, mean-score and Bayesian score) **w.r.t the number of tournament layers (which is one core design of the SynthID-Text watermarking system)**.
>
> Our **theoretical analysis** reveals
>
> - SynthID with mean-score has a fundamental vulnerability (Theorems 7): as the number of layers increases, regardless of the g-value distribution (Theorem 3-6), the detectability of the Mean-score progressively degrades. This insight motivates our proposed Layer Inflation Attack, which exploits this weakness by artificially increasing the number of layers to effectively remove the SynthID watermark.
>
> - SynthID with Bayesian-score is robust (Theorems 11) wrt increased number of layers, regardless of the g-value distribution (Theorem 8-10).
>
> The theoretical foundation for these observations are also validated through the experiments in Sec 5.
>
> ### **Q2: Able to use numerical plots corresponding to Theorems 7 and 11 to confirm that the theoretical trends match the empirical TPR curves reported in Section 5?**
>
> The theoretical TPR depends on the prompt distribution and LLM model and its true value cannot be calculated. We can only compute the empirical TPR on a particular dataset with certain prompts.

---

### Meta-Review · Area_Chair_3u6A · 2026-01-09

**Summary:**

This paper presents a theoretical analysis of SynthID-Text, a watermarking system deployed at Google. The reviews were split, with Xzg7 (8) and 3TGv (6) leaning toward acceptance and 9E5t (4) and nxDi (4) leaning toward rejection. A central theme across three reviews (3TGv, 9E5t, and nxDi) is that the paper’s scope is narrow: it focuses solely on SynthID-Text (and in places specifically the distortion-free variant) rather than offering more general insights about engineering "better" LLM watermarks or benchmarking against alternative watermarking approaches. This point was also acknowledged in a milder form by the most positive reviewer (Xzg7).

My own reading of the paper yields a tepid assessment consistent with 9E5t, nxDi, and 3TGv: the very limited scope, paired with a relatively shallow experimental setup in the original submission, makes the impact of this work less clear. There is certainly value in analyzing a widely used watermark. However, I also agree with 9E5t’s higher-level concern that the theoretical contributions largely stop short of translating into new and improved/generalized detection strategies or broader engineering insights beyond SynthID-Text.

**Reviewer Concerns:**

A major outstanding concern is scope/generalizability beyond SynthID-Text, raised explicitly by multiple reviewers. 3TGv asks: *“What is the rationale for focusing exclusively on Google's SynthID-Text method rather than conducting comparative analysis across multiple watermarking approaches?* This is an important question that remains unanswered post-rebuttal. Reviewer nxDi similarly frames the contribution as narrow. 9E5t also emphasizes this limitation, noting that *"The analysis focuses exclusively on the non-distortionary version of SynthID-Text, leaving the distortionary setting unexamined.”* I did not find the authors rebuttal that SynthID *"is currently the only industrial-scale, production-ready watermarking system deployed in real-world applications (by Google)"* convincing. Sure, this makes the results relevant to Google -- but why is it a contribution to the broader research on engineering better watermarks? Why is this of interest to the broader watermarking research community? This is **not** to say that the work is not valuable -- my point is simply that the authors must better position their results as a contribution to a broader literature and engage with a more extensive body of work on LLM watermarking. As it stands, the rebuttal reads as "it is important because Google uses it."

A second major concern is limited experimental validation in the original manuscript. 9E5t states: *“The experimental evaluation is limited to a single model (Gemma-7B-IT) and one dataset (ELI5).”* In their rebuttal, the authors added a table with additional results. But (as I see it) this does not fully resolve the broader benchmarking concern raised in the reviews, particularly relative to the breadth of evaluation now common in recent watermarking papers (see, for instance, https://arxiv.org/html/2506.06409v1 at the latest NeurIPS).

Finally, reviewers raised issues with the paper's clarity and assumptions, such as nxDi questioning the independence assumptions underlying parts of the CLT-based analysis. The authors did engage with nxDi post-rebuttal and addressed the clarification request, but nxDi still recommended that the clarifications be made explicit in the paper and advocated for making the manuscript more accessible to a general audience.

Overall, while the rebuttal adds experiments and clarifications, my view is that several concerns are outstanding: the scope and impact of the results beyond SynthID-Text, broader comparisons/benchmarks, and clearer treatment of assumptions.

**Reviewer Scores:**

Since only one reviewer engaged with the authors post-rebuttal, assessing the score change is more challenging in this case.

* 9E5t (4): I am hard-pressed to believe 9E5t would increase their score, as their concerns are largely fundamental (scope, limited benchmarking, and limited broader watermarking insight) rather than missing clarifications.
* nxDi (4): nxDi participated via a follow-up clarification request, but gave no indication they would change their score. I would thus expect the score to remain around 4.
* 3TGv (6): 3TGv was mildly positive but also highlighted scope/comparative-analysis gaps. I do not expect the rebuttal alone would make them raise their score above 6.
* Xzg7 (8): Xzg7 was strongly positive and already recommended acceptance. I do not expect a change upward given that even this review acknowledged the limited scope.

---

### Decision · Program_Chairs · 2026-01-26

Reject